# ICLR: In-Context Learning of Representations

**Core Francisco Park**[*1,2,3] , **Andrew Lee**[*4], **Ekdeep Singh Lubana**[*1,3], **Yongyi Yang**[*1,3,5],
**Maya Okawa**[1,3], **Kento Nishi**[1,4], **Martin Wattenberg**[4], **& Hidenori Tanaka**[1,3]

[1]CBS-NTT Program in Physics of Intelligence, Harvard University
[2]Department of Physics, Harvard University
[3]Physics & Informatics Lab, NTT Research Inc.
[4]SEAS, Harvard University
[5]CSE, University of Michigan, Ann Arbor

## Abstract

Recent work has demonstrated that semantics specified by pretraining data influence how representations of different concepts are organized in a large language model (LLM). However, given the open-ended nature of LLMs, e.g., their ability to in-context learn, we can ask whether models alter these pretraining semantics to adopt alternative, context-specified ones. Specifically, if we provide in-context exemplars wherein a concept plays a different role than what the pretraining data suggests, do models reorganize their representations in accordance with these novel semantics? To answer this question, we take inspiration from the theory of *conceptual role semantics* and define a toy "graph tracing" task wherein the nodes of the graph are referenced via concepts seen during training (e.g., `apple`, `bird`, etc.) and the connectivity of the graph is defined via some predefined structure (e.g., a square grid). Given exemplars that indicate traces of random walks on the graph, we analyze intermediate representations of the model and find that *as the amount of context is scaled, there is a sudden re-organization from pretrained semantic representations to* **in-context representations** *aligned with the graph structure.* Further, we find that when reference concepts have correlations in their semantics (e.g., `Monday`, `Tuesday`, etc.), the context-specified graph structure is still present in the representations, but is unable to dominate the pretrained structure. To explain these results, we analogize our task to energy minimization for a predefined graph topology, providing evidence towards an implicit optimization process to infer context-specified semantics. Overall, our findings indicate scaling context-size can flexibly re-organize model representations, possibly unlocking novel capabilities.

## 1 Introduction

A growing line of work demonstrates that large language models (LLMs) organize representations of specific concepts in a manner that reflects their structure in pretraining data (Park et al., 2024c;d; Engels et al., 2024; Abdou et al., 2021; Patel & Pavlick, 2022; Anthropic AI, 2024; Gurnee & Tegmark, 2023; Vafa et al., 2024; Li et al., 2021; Pennington et al., 2014). More targeted experiments in synthetic domains have further corroborated these findings, showing how model representations are organized according to the data-generating process (Li et al., 2022; Jenner et al., 2024; Traylor et al., 2022; Liu et al., 2022b; Shai et al., 2024; Park et al., 2024b; Gopalani et al., 2024). However, when a model is deployed in open-ended environments, we can expect it to encounter novel semantics for a concept that it did not see during pretraining. For example, assume that we describe to an LLM that a new product called `strawberry` has been announced. Ideally, based on this context, the model would alter the representation for `strawberry` and reflect that we are not referring to the pretraining semantics (e.g., the fruit strawberry). Does this ideal solution transpire in LLMs?

Motivated by the above, we evaluate whether when provided an in-context specification of a concept, an LLM alters its representations to reflect the context-specified semantics. Specifically, we propose

---

*Equal contribution. Contact: {corefranciscopark,andrewlee}@g.harvard.edu, yongyi@umich.edu, {ekdeeplubana, hidenori_tanaka}@fas.harvard.edu.

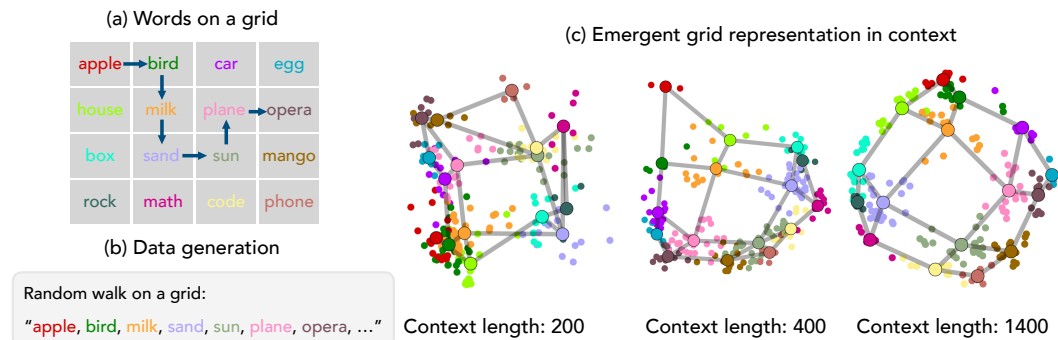

Figure 1: **Alteration of representations in accordance with context-specified semantics (grid structure).** (a) We randomly arrange a set of concepts on a grid that does not reflect any correlational semantics between the tokens. (b) We then generate sequences of tokens following a random walk on the grid, inputting it as context to a Llama-3.1-8B model. (c) The model's mean token representations projected onto the top two principal components. As the number of in-context exemplars increases, there is a formation of representations mirroring the grid structure underlying the data-generating process. Representations are from the residual stream activation following layer 26.

an in-context learning task that involves a simple "graph tracing" problem wherein the model is shown edges corresponding to a random traversal of a graph (see Fig. 1). The nodes of this graph are intentionally referenced via concepts the model is extremely likely to have seen during training (e.g., apple, bird, etc.), while its connectivity structure is defined using a predefined geometry that is ambivalent to correlations between concepts' semantics (e.g., a square grid). Based on the provided context, the model is expected to output a valid next node prediction, i.e., a node connected to the last presented one. *As we show, increasing the amount of context leads to a sudden re-organization of representations in accordance with the graph's connectivity.* This suggests LLMs can manipulate their representations in order to reflect concept semantics specified *entirely in-context*, inline with theories of inferential semantics from cognitive science (Harman, 1982; Block, 1998). We further characterize these results by analyzing the problem of Dirichlet energy minimization, showing that models indeed identify the structure of the underlying graph to achieve a non-trivial accuracy on our task. This suggests an implicit optimization process, as hypothesized by theoretical work on ICL in toy setups (e.g., in-context linear regression), can transpire in more naturalistic settings (Von Oswald et al., 2023a;b; Akyürek et al., 2023). Overall, our contributions can be summarized as follows.

- **Graph Navigation as a Simplistic Model of Novel Semantics.** We introduce a toy graph navigation task that requires a model to interpret semantically meaningful concepts as referents for nodes in a structurally constrained graph. Inputting traces of random walks on this graph into an LLM, we analyze whether the model alters its intermediate representations for referent concepts to predict valid next nodes as defined by the underlying graph connectivity, hence inferring, inline with theories of semantics from cognitive science, novel semantics of a concept (Harman, 1982).

- **Emergent In-Context Reorganization of Concept Representations.** Our results show that as context-size is scaled, i.e., as we add more exemplars in context, there is a sudden re-organization of concept representations that reflects the graph's connectivity structure. Intriguingly, these results are similar to ones achieved in a similar setup with human subjects (Garvert et al., 2017; Whittington et al., 2020). Further, we show the context-specified graph structure emerges even when we use concepts that have correlations in their semantics (e.g., Mon, Tues, etc.), but, interestingly, is unable to dominate the pretrained structure. More broadly, we note that this sudden reorganization is reminiscent of emergent capabilities in LLMs when other relevant axes, e.g., compute or model size, are scaled (Wei et al., 2022; Srivastava et al., 2022; Lubana et al., 2024).

- **An Energy Minimization Model of Semantics Inference.** To provide a more quantitative account of our results, we compute the Dirichlet energy of model representations with respect to the ground-truth graph structure, and find the energy decreases as a function of context size. This offers a precise hypothesis for the mechanism employed by an LLM to re-organize representations according to the context-specified semantics of a concept. These results also serve as

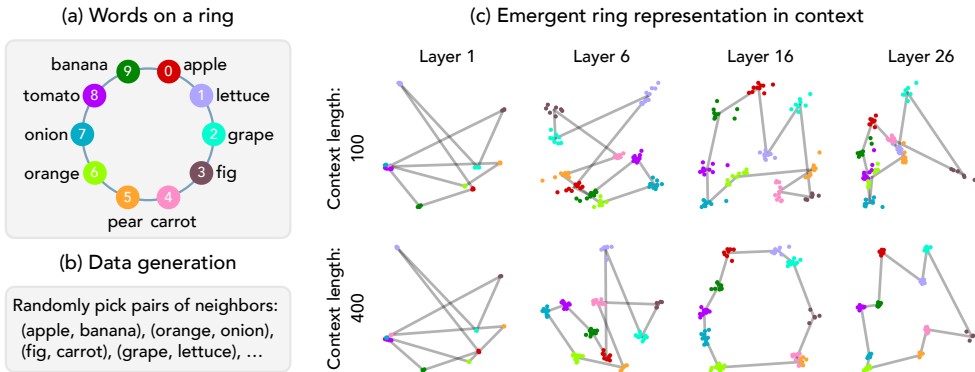

Figure 2: **Alteration of representations in accordance with context-specified semantics (ring structure).** (a) We randomly place concepts on a ring structure unrelated to their semantics. (b) We then generate sequences of tokens by randomly sampling *neighboring pairs* from the ring which is used as the input context to a Llama-3.1-8B model. (c) The model's mean representation of tokens projected onto the top two principal components. As the number of in-context exemplars increases, there is a formation of representations mirroring the ring structure underlying the data-generating process. The representations are from the residual stream activations.

evidence towards theories of in-context learning as implicit optimization in a more naturalistic setting (Von Oswald et al., 2023a;b; Akyürek et al., 2023).

## 2 EXPERIMENTAL SETUP: IN-CONTEXT GRAPH TRACING

We first define our setup for assessing the impact of context specification on how a model organizes its representations. In the main paper, we primarily focus on Llama3.1-8B (henceforth Llama3) (Dubey et al., 2024), accessed via NDIF/NNsight (Fiotto-Kaufman et al., 2024). We present results on other models—Llama3.2-1B / Llama3.1-8B-Instruct (Dubey et al., 2024) and Gemma-2-2B / Gemma-2-9B (Gemma Team, 2024)—in App. C.2.

**Task.** Our proposed task, which we call *in-context graph tracing*, involves random walks on a predefined graph $\mathcal{G}$. Specifically, inspired by prior work analyzing structured representations learned by sequence models, we experiment with three graphical structures: a square grid (Fig. 1 (a)), a ring (Fig. 2 (a)), and a hexagonal grid (Fig. 10). Results on hexagonal grid are deferred to appendix due to space constraints. To construct the square grid, we randomly arrange the set of tokens in a grid and add edges between horizontal and vertical neighbors. We then perform a random walk on the graph, emitting the visited tokens as a sequence (Fig. 1 (b)). For the ring, we add edges between neighboring nodes and simply sample random pairs of neighboring tokens on the graph (Fig. 2 (b)). Nodes in our graphs, denoted $\mathcal{T} = \{\tau_0, \tau_1, \ldots, \tau_n\}$, are referenced via concepts that the model is extremely likely to have seen during pretraining. While any choice of concepts is plausible, we select random tokens that, unless mentioned otherwise, have no obvious semantic correlations with one another (e.g., apple, sand, math, etc.). However, these concepts have precise meanings associated with them in the training data, necessitating that to the extent the model relies on the provided context, the representations are morphed according to the in-context graph. We highlight that a visual analog of our task, wherein one uses images instead of text tokens to represent a concept, has been used to elicit *very similar results with human subjects as the ones we report in this paper using LLMs* (Garvert et al., 2017; Whittington et al., 2020; Mark et al., 2020; 2024; Brady et al., 2009). We also note that our proposed task is similar to ones studied in literature on in-context RL, wherein one provides exploration trajectories in-context to a model and expects it to understand the environment and its dynamics (a.k.a., a world model) (Lee et al., 2024b; Laskin et al., 2022).

## 3 RESULTS

### 3.1 VISUALIZING INTERNAL ACTIVATION USING PRINCIPAL COMPONENTS

Since we are interested in uncovering context-specific representations, we input sequences from our data-generating process to the model and first compute the mean activations for each unique token $\tau \in \mathcal{T}$. Namely, assume a given context $\mathcal{C} := [c_0, ..., c_{N-1}]$, where $c_i \in \mathcal{T}$, that originates from an underlying graph $\mathcal{G}$. At each timestep, we look at a window of $N_w$ (=50) preceding tokens (or all tokens if the context length is smaller than $N_w$), and collect all activations corresponding to each token $\tau \in \mathcal{T}$ at a given layer $\ell$. We then compute the mean activations per token, denoted as $h_\tau^\ell \in \mathbb{R}^d$. We further denote the stack of mean token representations as $\boldsymbol{H}^\ell(\mathcal{T}) \in \mathbb{R}^{n \times d}$. Finally, we run PCA on $\boldsymbol{H}^\ell(\mathcal{T})$, and use the first two principal components to visualize model activations (unless stated otherwise). We note that while PCA visualizations are known to suffer from pitfalls as a representation analysis method, we provide a thorough quantitative analysis in Sec. 4 to demonstrate that the model re-organizes concept representations according to the in-context graph structure, and prove in Sec. 5 that the structure of the graph is reflected in the PCA visualizations *because* of this re-organization of representations. We also provide further evidence on the faithfulness of PCA by conducting a preliminary causal analysis of the principal components, finding that intervening on concept representations' projections along these components affects the model's ability to accurately predict valid next node generations (App. C.4).

**Results.** Figs. 1, 2 demonstrate the resulting visualizations for square grid and ring graphs, respectively (more examples are provided in the Appendix; see Fig. 9, 10). Strikingly, with enough exemplars, we find representations are in fact organized in accordance with the graph structure underlying the context. Interestingly, results can be skewed in the earlier layers towards semantic priors the model may have internalized during training; however, these priors are overridden as we go deeper in the model. For example, in the ring graph (see Fig. 2), concepts `apple` and `orange` are closer to each other in Layer 6 of the model, but become essentially antipodal around layer 26, as dictated by the graph; the antipodal nature is also more prominent as context length is increased.

We also observe that despite developing a square-grid structure when sufficient context length is given (see Fig. 1), the structure is partially irregular; e.g., it is wider in the central regions, but narrowly arranged in the periphery. We find this to be an artifact of frequency with which a concept is seen in the context. Specifically, due to lack of periodic boundary conditions, concepts that are present in the inner 2×2 region of the grid are visited more frequently during a random walk on the graph, while the periphery of the graph has a lower visitation frequency. The representations reflect this, thus organizing in accordance with both structure and frequency of concepts in the context.

Overall, the results above indicate that *as we scale context size, models can re-organize semantically unrelated concepts to form task-specific representations, which we call **in-context representations**.* Intriguingly, these results are broadly inline with theories of inferential semantics from cognitive science as well (Harman, 1982; Block, 1998).

### 3.2 SEMANTIC PRIOR VS. IN-CONTEXT TASK REPRESENTATIONS

Building on results from the previous section, we now investigate the impact of using semantically correlated concepts. Specifically, we build on the results from Engels et al. (2024), who show that representations for days of the week, i.e., {Monday, Tuesday, Wednesday, Thursday, Friday, Saturday, Sunday}, organize in a circular geometry. We randomly permute the ordering of these concepts, arrange them on a 7-node ring graph similar to the previous section (see Fig. 3a), and evaluate whether the in-context representations can override the strong pretraining prior internalized by the model.

**Results.** Fig. 3 (b, c) demonstrate the resulting visualizations. We find that when there is a conflict between the semantic prior and in-context task, we observe the *original semantic ring* in the first two principal components. However, the components right after in fact encode the context-specific structure: visualizing the third and fourth principal components shows the newly defined ring structure. This indicates that the context-specified structure is present in the representations, but does not dominate them. In Fig. 14, we report the model's accuracy on the in-context task, finding that the model overrides the semantic prior to perform well on our task when enough context is given.

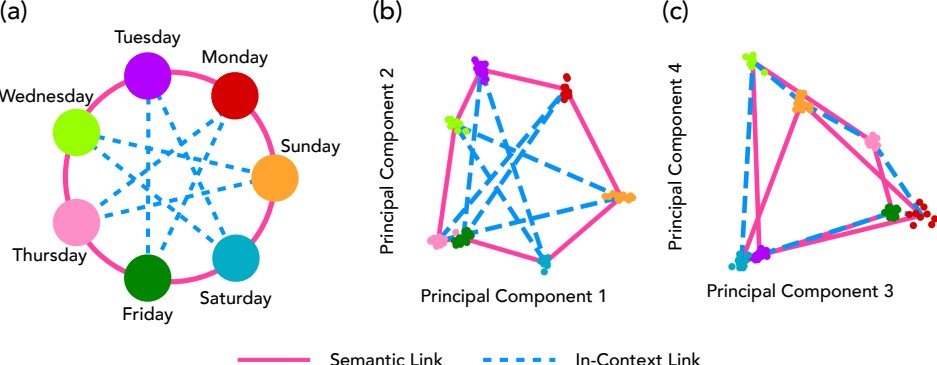

Figure 3: **In-context representations form in higher principal components in the presence of semantic priors.** (a) (Purple) Semantic links underlying days of the week. (Dashed blue) We define a non-semantic graph structure by linking non-neighboring days and generate tokens from this graph. (b) (Purple) The ring geometry formed by semantic links established during pre-training remains intact in the first two principal components. (c) (Dashed blue) The non-semantic structure provided in-context can be seen in the third and fourth principal components. Note that the star structure in the first two components (b), which match the ground truth graphical structure of our data generating process (a), becomes a ring in the next two principal components (c). The representations are from the residual stream activation following layer 21.

## 4 EFFECTS OF CONTEXT SCALING: EMERGENT RE-ORGANIZATION OF REPRESENTATIONS

Our results in the previous section demonstrate models can re-organize concept representations in accordance with the context-specified semantics. We next aim to study how this behavior arises as context is scaled—is there a continuous, monotonic improvement towards the context-specified structure as context is added? If so, is there a trivial solution, e.g., regurgitation based on context that helps explain these results? To analyze these questions, we must first define a metric that helps us gauge how aligned the representations are with the structure of the graph that underlies the context.

**Dirichlet Energy.** We measure the *Dirichlet energy* of our graph $\mathcal{G}$'s structure by defining an energy function over the model representations. Specifically, for an undirected graph $\mathcal{G}$ with $n$ nodes, let $\boldsymbol{A} \in \mathbb{R}^{n \times n}$ be its adjacency matrix, and $\boldsymbol{x} \in \mathbb{R}^n$ be a signal vector that assigns a value $x_i$ to each node $i$. Then the Dirichlet energy of the graph with respect to $\boldsymbol{x}$ is defined as

$$E_{\mathcal{G}}(\boldsymbol{x}) = \sum_{i,j} \boldsymbol{A}_{i,j}(x_i - x_j)^2. \tag{1}$$

For a multi-dimensional signal, the Dirichlet energy is defined as the summation of the energy over each dimension. Specifically, let $\boldsymbol{X} \in \mathbb{R}^{n \times d}$ be a matrix that assigns each node $i$ with a $d$-dimensional vector $\boldsymbol{x}_i$, then the Dirichlet energy of $\boldsymbol{X}$ is defined by

$$E_{\mathcal{G}}(\boldsymbol{X}) = \sum_{k=1}^{d} \sum_{i,j} \boldsymbol{A}_{i,j}(x_{i,k} - x_{j,k})^2 = \sum_{i,j} \boldsymbol{A}_{i,j}\|\boldsymbol{x}_i - \boldsymbol{x}_j\|^2. \tag{2}$$

Overall, to empirically quantify the formation of geometric representations, we can measure the Dirichlet energy with respect to the graphs underlying our data generating processes (DGPs) and our mean token activations $\boldsymbol{h}_\tau^\ell$:

$$E_{\mathcal{G}}(\boldsymbol{H}^\ell(\mathcal{T})) = \sum_{i,j} \boldsymbol{A}_{i,j}\|\boldsymbol{h}_i^\ell - \boldsymbol{h}_j^\ell\|^2, \tag{3}$$

where $\boldsymbol{H}^\ell(\mathcal{T}) \in \mathbb{R}^{n \times d}$ is the stack of our mean token representations $\boldsymbol{h}^\ell$ at layer $\ell$ and $i, j \in \mathcal{T}$ are tokens from our DGP at a certain context length. We note $\boldsymbol{H}^\ell(\mathcal{T})$ is a function of context

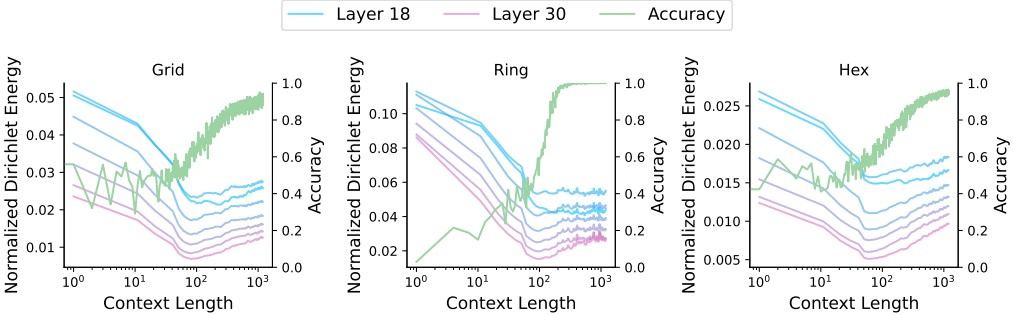

Figure 4: **A model continuously develops task representation as it learns to traverse novel graphs in-context.** We plot the accuracy of graph traversal and the Dirichlet energy of the graph, computed from the model's internal representations, as functions of context length. We note that the Dirichlet energy never reaches a perfect zero—ruling out that the representations are learning a degenerate structure, as was also seen in the PCA visualizations in Sec. 3. (a) A 4x4 grid graph with 16 nodes. (b) A circular ring with 10 nodes. (c) A "honey-comb" hexagonal lattice, with 30 nodes.

length as well, but we omit it in the notation for brevity. Intuitively, the measure above indicates whether neighboring tokens (nodes) in the ground truth graph have a small distance between their representations. *Thus, as the model correctly infers the correct underlying structure, we expect to see a decrease in Dirichlet energy.* We do note that, in practice, Dirichlet energy minimization has a trivial solution where all nodes are assigned the same representation. While we can be confident this trivial solution does not exist in our results, for else we would not see distinct node representations in PCA visualizations nor high accuracy for solving our tasks, we still provide an alternative analysis in App. C.3 where the representations are standardized (mean-centered and normalized by variance) to render this trivial solution infeasible. We find results are qualitatively similar with such standardized representations, but more noisy since standardization can induce sensitivity to noise.

## 4.1 RESULTS: EMERGENT ORGANIZATION AND TASK ACCURACY IMPROVEMENTS

We plot Llama3's accuracy at the in-context graph tracing task alongside the Dirichlet energy measure (for different layers) as a function of context. Specifically, we compute the "rule following accuracy", where we add up the model's output probability over all graph nodes which are valid neighbors. For instance, if the graph structure is `apple-car-bird-water` and the current state is `car`, we add up the predicted probabilities for `apple` and `bird`. This metric simply measures how well the model abides by the graph structure.

Results are reported in Fig. 4. We see once a critical amount of context is seen by the model, accuracy starts to rapidly improve. We find this point in fact closely matches when Dirichlet Energy reaches its minimum value: energy is minimized shortly before the rapid increase in in-context task accuracy, suggesting that the structure of the data is correctly learned before the model can make valid predictions. This leads us to the claim that *as the amount of context is scaled, there is an emergent re-organization of representations that allows the model to perform well on our in-context graph tracing task.* We note these results also provide a more quantitative counterpart of our PCA visualization results before.

**Is there a Trivial Solution at play?** A simple baseline that would exhibit an increase in performance with increasing context involves the model merely regurgitating a node's neighbors by copying them from its context. We call this the *memorization solution*. While such a solution would not explain the reorganization of representations, we use it as a baseline to show the model is likely engaging in a more intriguing mechanism. Since our accuracy metric measures rule following, this memorization solution will achieve value 1 if the node has been observed in the context and 0 otherwise. Following our data sampling process then, if we simply choose an initial node at random with replacement, we can express the probability of a node existing in a context of length $l$ as:

$$p_{\text{seen1}}(\mathbf{x}) = 1 - \left(\frac{n-1}{n}\right)^l, \tag{4}$$

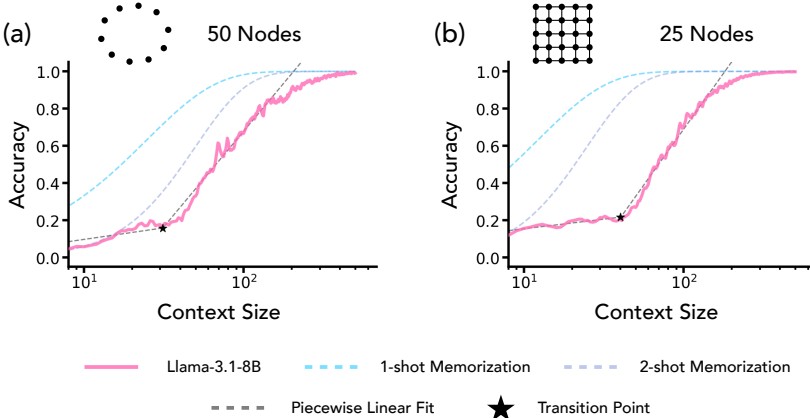

Figure 5: **A memorization solution cannot explain Llama's ICL graph tracing performance.**
We plot the rule-following accuracy from Llama-3.1-8B outputs and accuracies from a simple 1-shot
and 2-shot memorization hypothesis. (a) A ring graph with 50 nodes. (b) A square grid graph with
25 nodes. In both cases, we find that the memorization solution cannot explain the accuracy ascent
curve. Instead, we find a slow phase and a fast phase, which we fit with a piecewise linear fit.

where $\mathbf{x}$ is the context and $n$ is the number of nodes available. Note that the current node itself
does not matter as the sampling probability is uniform with replacement. We also evaluate another,
similar baseline that assumes the same token much be encountered twice for the model to recognize
it as an in-context exemplar. To define a closed-form expression for this solution, we have the
probability that a node has appeared twice as follows:

$$p_{\text{seen2}}(\mathbf{x}) = p_{\text{seen1}}(\mathbf{x}) - l \left( \frac{1}{n} \right)^1 \left( \frac{n-1}{n} \right)^{(l-1)}. \tag{5}$$

To evaluate whether the memorization solutions above explain our results, we plot their performance
alongside the observed performance of Llama-3. Fig. 5 shows the result (a) on a ring graph with 50
nodes and (b) on a grid graph with 25 nodes. We find, in both cases, that neither the 1-shot nor the
2-shot memorization curve can explain the behavior of Llama. Instead, we observe that the accuracy
has two phases, a first phase where the accuracy improves very slowly, and a second phase where
the log-linear slope suddenly changes to a steeper ascent. We find that a piecewise linear fit can
extract this transition point fairly well, which will be of interest in the next section.

## 5 EXPLAINING EMERGENT RE-ORGANIZATION OF REPRESENTATIONS: THE ENERGY MINIMIZATION HYPOTHESIS

Building on the results from previous section, we now put forward a hypothesis for why we are able
to identify such structured representations from a model: we hypothesize the model internally runs
an *energy minimization process* in search of the correct structural representation of the data (Yang
et al., 2022), similar to claims of implicit optimization in in-context learning proposed by prior work
in toy settings (Von Oswald et al., 2023a;b). More formally, we claim the following hypothesis.

**Hypothesis 5.1.** *Let $n$ be the number of tokens, $d$ be the dimensionality of the representations, and
$\boldsymbol{H}^{(\ell,t)}(\mathcal{T}) \in \mathbb{R}^{n \times d}$ be the stack of representations for each token learned by the model at layer $\ell$
and context length $t$, then $E_{\mathcal{G}}\left(\boldsymbol{H}^{(\ell,t)}(\mathcal{T})\right)$ decays with context length $t$.*

### 5.1 MINIMIZERS OF DIRICHLET ENERGY AND SPECTRAL EMBEDDINGS.

We call the $k$-th energy minimizer of $E_{\mathcal{G}}$ the optimal solution that minimizes $E_{\mathcal{G}}$ and is orthogonal
to the first $k-1$ energy minimizers. Formally, the energy minimizers $\left\{ \boldsymbol{z}^{(k)} \right\}_{k=1}^{n}$ are defined as the

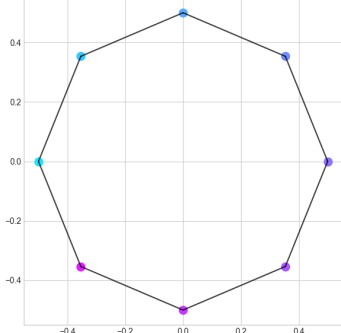 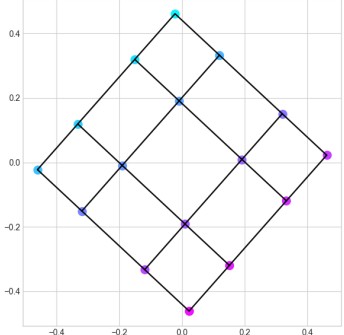

Figure 6: **Spectral embedding of a ring graph.**   Figure 7: **Spectral embedding of a grid graph.**

solution to the following problem:

$$\boldsymbol{z}^{(k)} = \arg \min_{\boldsymbol{z} \in \mathbb{S}^{n-1}} E_{\mathcal{G}}(\boldsymbol{z}) \tag{6}$$

$$\text{s.t.} \quad \boldsymbol{z} \perp \boldsymbol{z}^{(j)}, \forall j \leq k-1, \tag{7}$$

where $\mathbb{S}^{n-1}$ is the unit sphere in $n$ dimensional Euclidean space. The energy minimizers are known to have the following properties (Spielman, 2019):

1. $\boldsymbol{z}^{(1)} = c\boldsymbol{1}$ for some constant $c \neq 0$, which is a degenerated solution that assigns the same value to every node; and

2. If we use $\left(z_i^{(2)}, z_i^{(3)}\right)$ as the coordinate of node $i$, it will be a good planar embedding. We call them (2-dimensional) **spectral embeddings**.

Spectral embeddings are often used to a draw graph on a plane and in many cases can preserve the structure of the graph (Tutte, 1963). In Figs. 6 and 7, we show the spectral embedding results for a ring graph and a grid graph respectively. Notice how such spectral embeddings are similar to the representations from our models in Fig. 1 and 2. *As we show in Theorem B.1, this is in fact expected if our energy minimization hypothesis is true:* if the representations $\boldsymbol{H}$ from the model minimize the Dirichlet energy and are non-degenerated, then the first two principal components of PCA will exactly produce the spectral embeddings $\boldsymbol{z}^{(2)}, \boldsymbol{z}^{(3)}$. Here we present an informal version of the theorem, and defer the full version and proof to the appendix.

**Theorem 5.1** (Informal Version of Theorem B.1)**.** *Let $\mathcal{G}$ be a graph and $\boldsymbol{H} \in \mathbb{R}^{n \times d}$ (where $n \geq d \geq 3$) be a matrix that minimizes Dirichlet energy on $\mathcal{G}$ with non-degenerated singular values, then the first two principal components of $\boldsymbol{H}$ will be $\boldsymbol{z}^{(2)}$ and $\boldsymbol{z}^{(3)}$.*

See App. B for the formal version and proof of Theorem 5.1. See also Tab. 2 for an empirical validation of the theorem, wherein we show the principal components align very well with spectral embeddings of the graph.

## 5.2 ENERGY MINIMIZATION AND GRAPH CONNECTIVITY

Given the relationship between spectral embeddings (i.e., energy minimizers) and the principal components observed in our results (Figs. 1, 2), we claim that the model's inference of the underlying structure is akin to an implicit energy minimization. To further analyze the implication of this claim, we show that the moment at which we can visualize a graph using PCA is the moment at which the model has found a large connected component (i.e., the graph's structure). Specifically, consider an unconnected graph $\hat{\mathcal{G}}$, i.e., $\hat{\mathcal{G}}$ has multiple connected components. Then, there are multiple degenerate solutions to the energy minimization problem, which will be found by PCA. Specifically, suppose $\hat{\mathcal{G}}$ has $q$ connected components, with $U_i$ denoting the set of nodes of the $i$-th component.

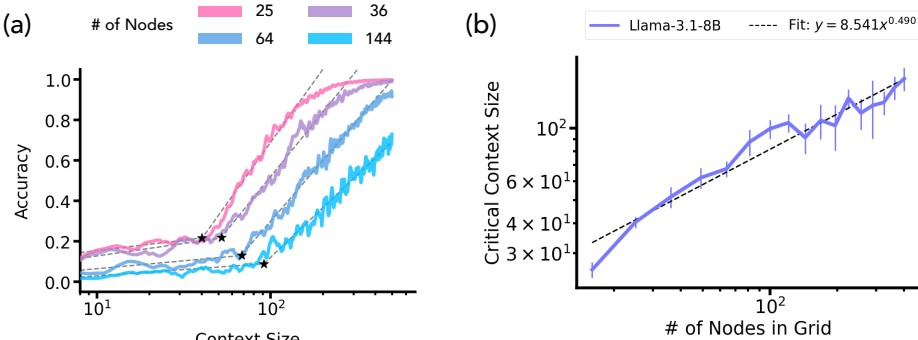

Figure 8: **In-context emergence.** We analyze the in-context accuracy curves as a function of context-size inputted to the model. The graph used in this experiment is an $m \times m$ grid, with a varying value for $m$. (a) The rule following accuracy of a graph tracing task. The accuracy show a two phase ascent. We fit a piecewise linear function to the observed ascent to extract the transition point, which moves rightwards with increasing graph size. (b) Interestingly, the transition point scales as a power-law in $m$, i.e., the number of nodes in the graph.

Then we can construct the first $q$ energy minimizers as follows: $\forall i \in [q]$, let the $j$-th value of $\boldsymbol{z}^{(i)}$ be

$$
z_j^{(i)} = \begin{cases} -\alpha_i & j \in \bigcup_{k=1}^{i-1} U_k \\ 1 & \text{otherwise,} \end{cases}
\tag{8}
$$

where $\alpha_1 = 1$ and $\alpha_i = \frac{\sum_{k'=i}^{q} \sum_{j \in U_{k'}} z_{j'}^{(i-1)}}{\sum_{k=1}^{i-1} \sum_{j \in U_k} z_j^{(i-1)}}$ for $i \in [q] \setminus \{1\}$.

It is easy to check that each $\boldsymbol{z}^{(i)}$ constructed above for $i \in [q]$ has 0 energy, and is thus a global minimizer of $E_{\hat{\mathcal{G}}}$. Moreover, all $\boldsymbol{z}^{(i)}$'s are orthogonal to each other and hence satisfy our definition of the first $q$ energy minimizers. It is important to notice that these $\boldsymbol{z}^{(i)}$'s for $i \in [q]$ contain no information about the structure of the graph, other than identifying each connected component. Theorem B.1 tells us that the principal components of a non-degenerated (rank $s$ where $s > 1$) solution $\boldsymbol{H}$ that minimizes the energy will be $\boldsymbol{z}^{(2)} \cdots \boldsymbol{z}^{(s+1)}$. *Thus, if the graph is unconnected, then the energy-minimizing representations will be dominated by information-less principal components, in which we should not expect any meaningful visualization.* The acute reader may recall that the first minimizer $\boldsymbol{z}^{(1)}$ is a trivial solution of the energy minimization that assigns the same value to every node. Conveniently, the above argument also implies that this is not a concern: PCA will rule out this degenerate solution as demonstrated in Theorem B.1.

**In-context emergence: A hypothesis.** Our results in Fig. 5 showed an intriguing breakpoint that is reminiscent of a second-order phase transition (i.e., an undefined second derivative). As shown in Fig. 8, we in fact find this behavior is extremely robust across graphs of different sizes, and shows a power-law scaling trend with increasing graph size (see App. C.7 for several more results in this vein, including different graph topologies). Given the relationship offered between energy minimization and discovery of a connected component (graph structure) in our analysis above, a possible framework to explain these results may be the problem of bond-percolation on a graph (Newman, 2003; Hooyberghs et al., 2010): in bond-percolation, one starts with an unconnected graph and slowly fills edges to connect its nodes; as edges are filled, there is a second-order transition after which a large connected component emerges in the graph. The nature of the transition observed in our experiments (Fig. 8) and the theoretical connection between energy minimization and existence of a connected component provide some evidence towards the plausibility of this hypothesis. However, we believe the analogy is still loose, for our graph sizes are relatively small (likely causing significant finite-size effects) and the experiments need to corroborate any scaling theory of the transition point from percolation literature would require running graphs with at least 2 orders-of-magnitude difference in their sizes. However, the consistency of the hypothesis with our empirical results and analysis implies that investigating it further may be fruitful.

## 6 RELATED WORK

**Model Representations.** Researchers have recently discovered numerous structured representations in neural networks. Mikolov et al. (2013) suggests that concepts are *linearly* represented in activations, and Park et al. (2024d) more recently suggests this may be the case for contemporary language models. Numerous researchers have found concrete examples of linear representations for human-level concepts, including "truthfulness" (Burns et al., 2022; Li et al., 2023b; Marks & Tegmark, 2024), "refusal" (Arditi et al., 2024), toxicity (Lee et al., 2024a), sycophancy (Rimsky et al., 2024), and even "world models" (Li et al., 2022; Nanda et al., 2023). Park et al. (2024c) finds that hierarchical concepts are represented with a tree-like structure consisting of orthogonal vectors. A relevant line of work includes that of Todd et al. (2023) and Hendel et al. (2023). Both papers find that one can compute a vector from in-context exemplars that encode the task, such that adding such a vector during test time for a new input can correctly solve the task. Language models do not always form linear representations, however. Engels et al. (2024) find circular feature representations for periodic concepts, such as days of the week or months of the year, using a combination of sparse autoencoders and PCA. Csordás et al. (2024) finds that recurrent neural networks trained on token repetition can either learn an "onion"-like representation or a linear representation, depending on the model's width. *Unlike such prior work, we find that task-specific representations with a desired structural pattern can be induced in-context.* To our knowledge, our work offers the first such investigation of in-context representation learning.

**Scaling In-Context Learning** Numerous works have demonstrated that in-context accuracy improves with more exemplars (Brown et al., 2020; Lu et al., 2022; Bigelow et al., 2023). With longer context lengths becoming available, researchers have begun to study the effect of *many-shot* prompting (as opposed to few-shot) (Agarwal et al., 2024; Anil et al., 2024; Li et al., 2023c). For instance, Agarwal et al. (2024) reports improved performance on ICL using hundreds to thousands of exemplars on a wide range of tasks. Similarly, Anil et al. (2024) demonstrate the ability to jail-break LLMs by scaling the number of exemplars. Unlike such work that evaluates model behavior, *we study the effect of scaling context on the underlying representations*, and provide a framework for predicting when discontinuous changes in behavior can be expected via mere context-scaling.

**Synthetic Data for Interpretability** Recent works have demonstrated the value of interpretable, synthetic data generating processes for understanding Transformer's behavior, including in-context learning (Park et al., 2024a; Ramesh et al., 2023; Garg et al., 2023), language acquisition (Lubana et al., 2024; Qin et al., 2024; Allen-Zhu & Li, 2023b), fine-tuning (Jain et al., 2023; Lubana et al., 2023; Juneja et al., 2022), reasoning abilities (Prystawski et al., 2024; Khona et al., 2024; Wen et al., 2024; Liu et al., 2022a), and knowledge representations (Nishi et al., 2024; Allen-Zhu & Li, 2023a). While prior work typically pre-trains Transformers on synthetic data, *we leverage synthetic data to study representation formation during in-context learning in pretrained large language models*.

## 7 DISCUSSION

In this work, we show that LLMs can flexibly manipulate their representations from semantics internalized based on pretraining data to semantics defined entirely in-context. To arrive at these results, we propose a simple but rich task of graph tracing, wherein traces of random walks on a graph are shown to the model in-context. The graphs are instantiated using predefined structures (e.g., lattices) and concepts that are semantically interesting (e.g., to define nodes), but meaningless in the overall context of the problem. Interestingly, we find the ability to flexibly manipulate representations is in fact emergent with respect to context size—we propose a model based on energy minimization to hypothesize a mechanism for the underlying dynamics of this behavior. These results suggest context-scaling can unlock new capabilities, and, more broadly, this axis may have as of yet been underappreciated for improving a model. *In fact, we note that, to our knowledge, our work is to first to investigate the formation of representations entirely in-context.* Our study also naturally motivates future work towards formation of world representations Li et al. (2023a) and world models (Ha & Schmidhuber, 2018) in-context, which can have significant implications toward building general and open-ended systems, as well as forecasting its safety concerns. We also highlight the relation of our experimental setup to similar tasks studied in neuroscience literature Garvert et al. (2017); Mark et al. (2020; 2024), wherein humans are shown random walks of a graph of visual concepts; fMRI images of these subjects demonstrate the formation of a structured representation of the graph in the hippocampal–entorhinal cortex, similar to our results with LLMs.

## ACKNOWLEDGMENTS

We greatly thank the National Deep Inference Fabric (NDIF) pilot program (Fiotto-Kaufman et al., 2024), especially, Emma Bortz, Jaden Fiotto-Kaufman, Adam Belfki, David Bau, and the team who provided us with access to representations of Llama models, making this work possible. CFP and HT acknowledge the support of Aravinthan D.T. Samuel, Cecilia Garraffo and Douglas P. Finkbeiner. CFP, ESL, KN, MO, and HT are supported by the CBS-NTT Program in Physics of Intelligence. AL and MW acknowledges support from the Superalignment Fast Grant from OpenAI. MW also acknowledges support from Effective Ventures Foundation, Effektiv Spenden Schweiz, and the Open Philanthropy Project. Part of the computations in this paper were run on the FASRC cluster supported by the FAS Division of Science Research Computing Group at Harvard University. ESL thanks Eric Bigelow for crucial discussions that helped define several hypotheses pursued in this work, and the Harvard CoCoDev lab (especially Peng Qian) and Talia Konkle for feedback on an earlier version of the paper. CFP thanks Zechen Zhang, Eric Todd, Clement Dumas, and Shivam Raval for useful discussions. All authors thank David Bau and his lab for useful feedback on the paper's results.

## AUTHOR CONTRIBUTIONS

CFP, AL, ESL, and HT conceived the in-context graph traversal task, inspired by discussions and experimentations with MO and KN in a related work on pretraining. CFP and AL co-led experiments, where CFP discovered the in-context learning of the ring-structured representation with input from HT, kicking off this study. AL suggested extending this to grid structures to CFP connecting it to world representations, and ESL proposed hexagonal configurations. ESL hypothesized in-context transitions with increased context and percolation mechanism. CFP conducted semantic overriding and percolation experiments, while AL performed accuracy-energy experiments and causal interventions, with CFP, AL, and ESL developing transition point detection methods and contributing to energy normalization. YY formulated the energy minimization theory and its proofs, collaborating with ESL to connect the framework to component existence and demonstrate PCA's optimality. MW provided alternative hypotheses for experimental robustness. CFP, AL, ESL, YY, and HT co-developed an initial manuscript with feedback from MW, with ESL leading the final writing and project narrative development. CFP, AL, and HT created figures, while CFP and AL jointly developed the appendix. AL conducted substantial experiments to verify the main claims generalize to other models with CFP's support. HT supervised the project.

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

## A   ADDITIONAL EXPERIMENTAL DETAILS

Here we provide some additional details regarding our experimental setups.

**Context Windows.**   Our analyses require computing mean token representations $h_i$ for every token $i \in \mathcal{T}$ in our graphs. To do so, we grab the activations per each token in the most recent context window of $N_w$ tokens. Because we further require that each token is observed at least once in our window, we use a batch of prompts, where the batch size is equal to the number of nodes in our graph. For each prompt in the batch, we start our random traversal (or random pairwise sampling) with a different node, ensuring that each node shows up at least once in the context. In the case when our context length ($N_c$) is longer than the window, we simply use every token ($N_w = N_c$).

**Computational Resources.**   We run our experiments on either A100 nodes, or by using the APIs provided by NDIF (Fiotto-Kaufman et al., 2024).

## B   THE CONNECTION BETWEEN ENERGY MINIMIZATION AND PCA STUCTURE

In this section, for a matrix $M \in \mathbb{R}^{n \times d}$, we use lowercase bold letters with subscript to represent the columns for $M$, e.g. $m_k$ represents the $k$-th column of $M$. Moreover, we use $\sigma_k(M)$ to represent the $k$-th largest singular value of $M$ and when $M$ is PSD we use $\lambda_k(M)$ to represent the $k$-th largest eigenvalue of $M$. Moreover, we use $e_k$ to represent a vector with all-zero entries except a 1 at entry $k$, whose dimension is inferred from context, and $\mathbf{1}$ to represent a vector with all entries being 1. For a natural number $n$, we use $[n]$ to represent $\{1, 2, \cdots, n\}$.

Furthermore, we use $\left\{ z^{(k)} \right\}_{k=1}^n$ to represent the energy minimizers of the Dirichlet energy, defined in Section 4. Let $A \in \mathbb{R}^{n \times n}$ be the adjacency matrix of the graph, $D = \mathrm{diag}(A\mathbf{1})$ be the degree matrix, and $L = D - A$ be the Laplacian matrix. Through an easy calculation one can know that for any vector $x \in \mathbb{R}^n$,

$$E_{\mathcal{G}}(x) = \langle x, Lx \rangle. \tag{9}$$

Therefore, from the Spectral Theorem (e.g. Theorem 2.2.1 in Spielman (2019)), we know that $z_k$ is the eigenvector of $L$ corresponding to $\lambda_{n-k+1}(L) = E_{\mathcal{G}}(z_k)$.

We will show that, if a matrix $H \in \mathbb{R}^{n \times d}$ minimizes the energy and is non-degenerated (has several distinct and non-zero singular values), then the PCA must exactly give the leading energy minimizers, starting from $z_2$.

**Theorem B.1.** *Let $\mathcal{G}$ be a graph and $\epsilon_1 > \epsilon_2 > \cdots > \epsilon_s > 0$ be $s \leq \min\{n, d\} - 1$ distinct positive numbers. Let matrix $H \in \mathbb{R}^{n \times d}$ be the solution of the following optimization problem:*

$$H = \arg \min_{X \in \mathbb{R}^{n \times d}} E_{\mathcal{G}}(X) \tag{10}$$

$$s.t. \quad \sigma_k(X) \geq \epsilon_k, \ \forall k \in [r], \tag{11}$$

*then the $k$-th principle component of $H$ (for $k \in [r]$) is $z^{(k+1)}$.*

*Proof.* We first prove that the leading left-singular vectors of $H$ are exactly energy minimizers. Let $r = \min\{n, d\}$. Let the SVD of $H$ be $H = U\Sigma V^\top$, where $\Sigma = \mathrm{diag}\left[\sigma_1, \sigma_2, \cdots, \sigma_d\right]$ are the singular values of $H$, and $U \in \mathbb{R}^{n \times r}$, $V \in \mathbb{R}^{r \times d}$.

Let $\boldsymbol{h}'_i$ represents the $i$-th row of $\boldsymbol{H}$. Notice that

$$E_{\mathcal{G}}(\boldsymbol{H}) = \sum_{i,j} \boldsymbol{A}_{i,j} \left\| \boldsymbol{h}'_i - \boldsymbol{h}'_j \right\|^2 \tag{12}$$

$$= \sum_{i,j} \boldsymbol{A}_{i,j} \left\| (\boldsymbol{e}_i - \boldsymbol{e}_j)^\top \boldsymbol{H} \right\|^2 \tag{13}$$

$$= \sum_{i,j} \boldsymbol{A}_{i,j} \left\| (\boldsymbol{e}_i - \boldsymbol{e}_j)^\top \boldsymbol{U}\boldsymbol{\Sigma} \right\|^2 \tag{14}$$

$$= \sum_{i,j} \sum_{k=1}^{r} \sigma_k^2 \left\langle \boldsymbol{e}_i - \boldsymbol{e}_j, \boldsymbol{u}_k \right\rangle^2 \tag{15}$$

$$= \sum_{k=1}^{r} \sigma_k^2 E_{\mathcal{G}}(\boldsymbol{u}_k). \tag{16}$$

Since $\sigma_k$'s and $\boldsymbol{u}_k$'s are independent, no matter what are the values of $\boldsymbol{u}_k$, we know that each $\sigma_k$ will take the smallest possible value, and from the given condition, it is $\sigma_k = \epsilon_k, \forall k \in [s]$, and $\sigma_k = 0, \forall k \in [r] \setminus [s]$.

Since $\boldsymbol{u}_k$'s are singular vectors, we have $\boldsymbol{u}_k$'s are orthogonal to each other. Using Theorem 1 in Fan (1949), we know that for any $s' \in [n]$, the minimizer of $\sum_{k=1}^{s'} E_{\mathcal{G}}(\boldsymbol{u}_k)$ is $\boldsymbol{u}_k = \boldsymbol{z}^{(k)}, \forall k \in [s']$. Therefore, it is evident that the minimizer of $\sum_{k=1}^{s} \sigma_k^2 E_{\mathcal{G}}(\boldsymbol{u}_k)$ must satisfies $\boldsymbol{u}_k = \boldsymbol{z}^{(k)}, \forall k \in [s]$, since from the above argument of $\sigma_k$'s and the given condition condition we know that $\sigma_1 > \sigma_2 > \cdots > \sigma_s > 0$.

Now we have proved that $\boldsymbol{u}_k = \boldsymbol{z}^{(k)}, \forall k \in [s]$. Next we consider the output of PCA. Let $\boldsymbol{p}_k$ be the $k$-th principle component output by the PCA of $\boldsymbol{H}$. We know that $\boldsymbol{p}_k$ is the eigenvector of

$$\boldsymbol{C} = \widehat{\boldsymbol{H}}\widehat{\boldsymbol{H}}^\top \tag{17}$$

that corresponds to the $k$-th largest eigenvalue of $\boldsymbol{C}$, where $\widehat{\boldsymbol{H}} = \boldsymbol{H} - \frac{1}{n}\boldsymbol{1}\boldsymbol{1}^\top\boldsymbol{H}$ is the centralized $\boldsymbol{H}$.

From the Spectral Theorem, we have

$$\boldsymbol{p}_k = \arg \max_{\substack{\boldsymbol{p} \in \mathbb{S}^{n-1} \\ \boldsymbol{p} \perp \boldsymbol{p}_i, \forall i \leq k-1}} \langle \boldsymbol{p}, \boldsymbol{C}\boldsymbol{p} \rangle. \tag{18}$$

Let $J = \operatorname{span}\{\boldsymbol{1}\}$ be the set of vectors whose every entry has the same value. Let $J^\perp$ be the subspace in $\mathbb{R}^n$ that is orthogonal to $J$. For a subspace $K$ of $\mathbb{R}^n$, let $\Pi_K : \mathbb{R}^n \to \mathbb{R}^n$ be the projection operator onto $K$.

We have that

$$\boldsymbol{p}_1 = \arg \max_{\boldsymbol{p} \in \mathbb{S}^{n-1}} \langle \boldsymbol{p}, \boldsymbol{C}\boldsymbol{p} \rangle \tag{19}$$

$$= \arg \max_{\boldsymbol{p} \in \mathbb{S}^{n-1}} \left\langle \boldsymbol{p}, \left(\boldsymbol{I} - \frac{1}{n}\boldsymbol{1}\boldsymbol{1}^\top\right) \boldsymbol{H}\boldsymbol{H}^\top \left(\boldsymbol{I} - \frac{1}{n}\boldsymbol{1}\boldsymbol{1}^\top\right) \boldsymbol{p} \right\rangle \tag{20}$$

$$= \arg \max_{\boldsymbol{p} \in \mathbb{S}^{n-1}} \left\langle \Pi_{J^\perp}(\boldsymbol{p}), \boldsymbol{H}\boldsymbol{H}^\top \Pi_{J^\perp}(\boldsymbol{p}) \right\rangle \tag{21}$$

$$= \arg \max_{\substack{\boldsymbol{p} \in \mathbb{S}^{n-1} \\ \boldsymbol{p} \perp J}} \left\langle \boldsymbol{p}, \boldsymbol{H}\boldsymbol{H}^\top \boldsymbol{p} \right\rangle, \tag{22}$$

which, again from Spectral Theorem, is the eigenvector of the second largest eigenvalue of $\boldsymbol{H}\boldsymbol{H}^\top$, which is $\boldsymbol{u}_2 = \boldsymbol{z}^{(2)}$. Using an induction and the same reasoning, it follows that for any $k \in [s]$, we have $\boldsymbol{p}_k = \boldsymbol{z}^{(k+1)}$. This proves the proposition. $\square$

# C ADDITIONAL RESULTS

## C.1 DETAILED LAYER-WISE VISUALIZATION OF REPRESENTATIONS

In Figure 9 and Figure 10 we provide additional visualizations per layer for each of our models and each of our data generating processes.

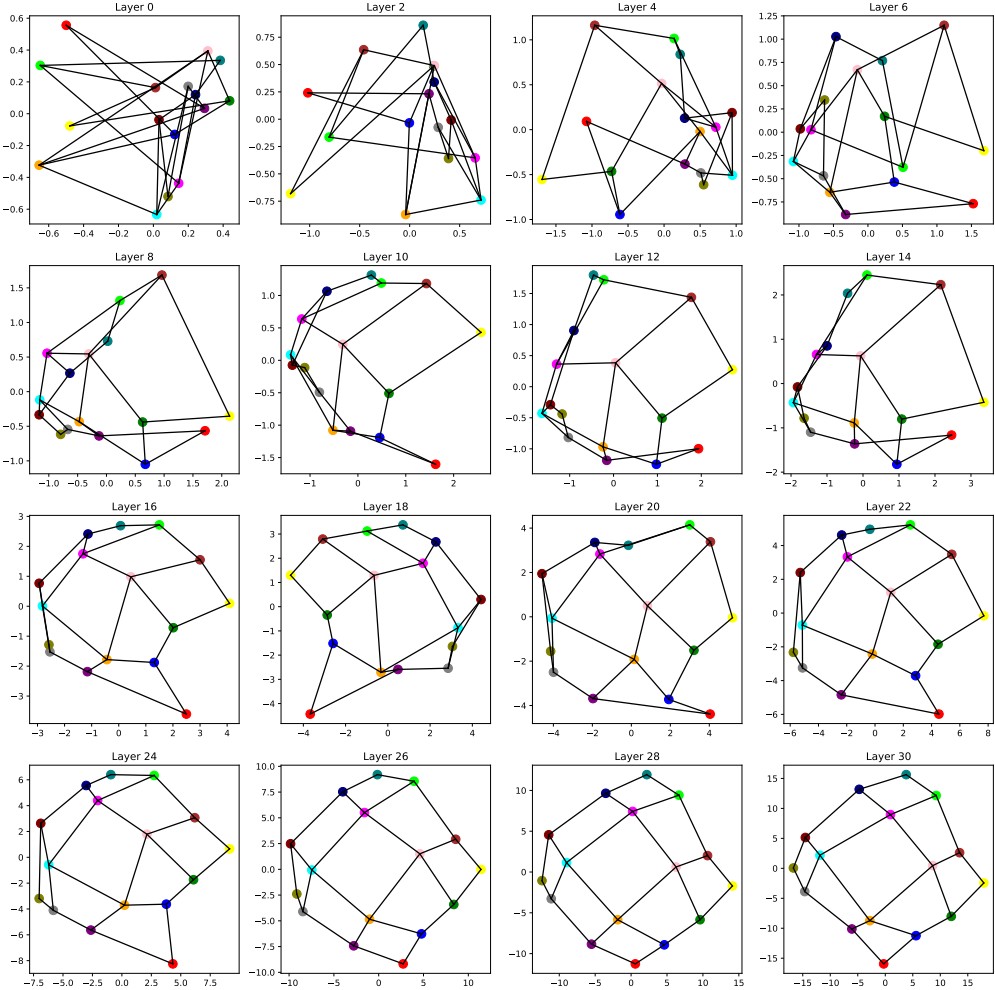

Figure 9: We plot 2d PCA projections from every other layer in Llama3.1-8B (Dubey et al., 2024), given the grid-traversal task. In deeper layers, we can see a clear visualization of the grid.

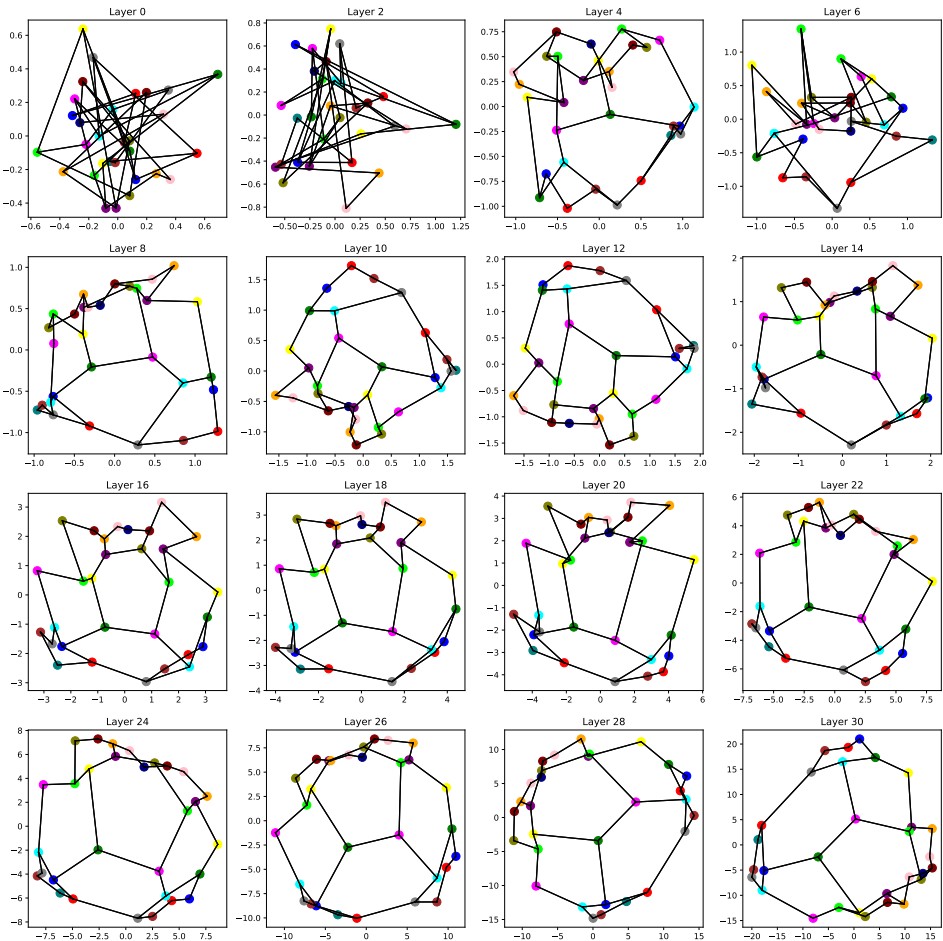

Figure 10: We plot 2D PCA projections from every other layer in Llama3.1-8B (Dubey et al., 2024) for the hexagonal grid task.

## C.2 PCA, Dirichlet Energy, and Accuracy Results on Other Models

Here we provide results from other language models, i.e., Llama3-1B (Dubey et al., 2024), Llama3-8B-Instruct, Gemma2-2B (Gemma Team, 2024), and Gemma2-9B. In Figure 11, we plot the 2d PCA projections from the last layer of various models for various data generating processes. In Figure 12, we plot the normalized Dirichlet energy curves against accuracy for various language models on various tasks. Across all models and tasks, we see results similar to the main paper.

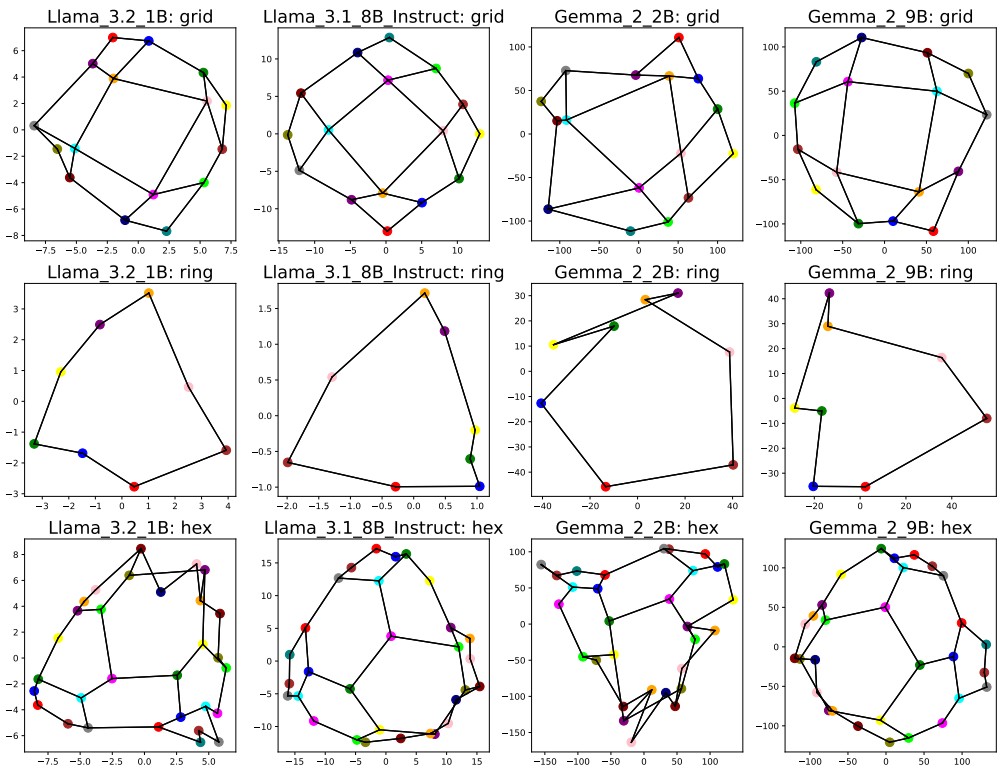

Figure 11: We plot 2d PCA projections from the last layer of various language models, given various data generating processes. For the grid and hexagonal graphs, we apply PCA on the last layers. For the rings, we visualize layers 14, 10, 16, and 20 respectively. Interestingly, for Llama3.2-1B, we find the ring representation in the 2nd and 3rd principal components.

## C.3 Standardized Dirichlet Energy

In Fig. 13, we report Dirichlet energy values computed after standardization of representations, i.e., after mean-centering them and normalizing by the standard deviation. This renders the trivial solution to Dirichlet energy minimization infeasible, since assigning a constant representation to all nodes will yield infinite energy (due to zero variance). As can be seen in our results, the plots are qualitatively similar to the non-standardized energy results (Fig. 12), but more noisy, especially for the ring graphs. This is expected, since standardization can exacerbate the influence of noise, yielding fluctuations in the energy calculation.

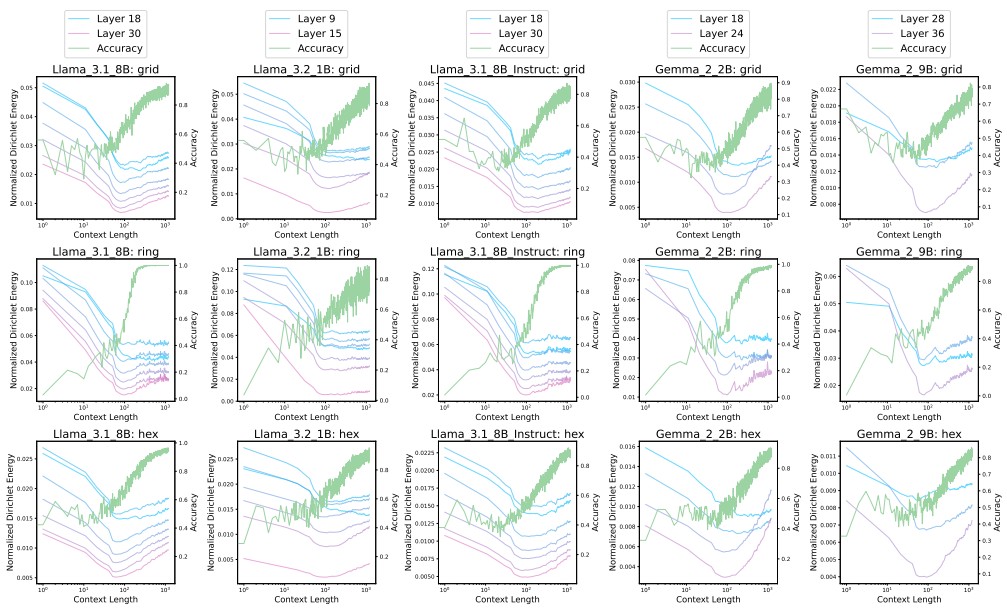

Figure 12: Accuracy versus normalized Dirichlet energy curves for various language models on various tasks. For every model and task, we see energy minimized before accuracy starting to improve.

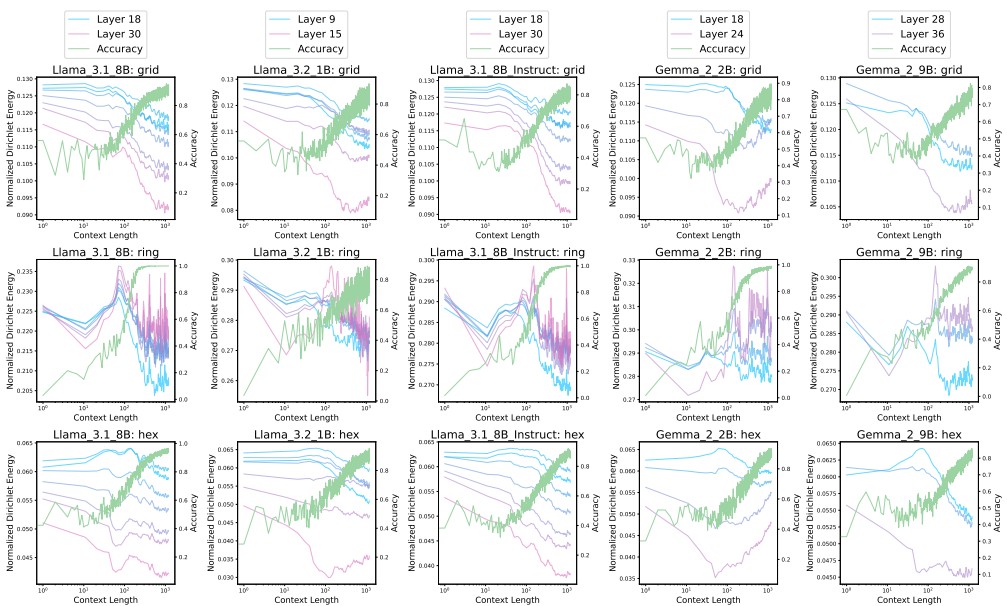

Figure 13: Accuracy versus zero mean centered normalized Dirichlet energy curves for various language models on various tasks. Zero mean centering ensures that graph representations are not using the trivial solution to energy minimization (i.e., assigning the same representation for every node).

## C.4 CAUSAL ANALYSIS OF REPRESENTATIONS

In this section we report preliminary causal analyses of our graph representations. While fully understanding the mechanisms behind the formation of such representations, as well as the relationship between said representations and model outputs are an interesting future direction, this is not the focus of our work and thus we only ran proof-of-concept experiments.

With that said, we ask: do the principal components that encode our graph representations have any causal role in the model's predictions?

To test this, we attempt to "move" the location of the activations for one node of the graph to another by simply re-scaling its principal components. Namely, assume activation $h_i^\ell$ corresponding to node $i$ at layer $\ell$. Say we wish to "move" the activation to a different *target node* $j$. We first compute the mean representation of node $j$ using all activations corresponding to node $j$ within the most recent $N_w$ (= 200) timesteps, notated as $\bar{h}_j$. Assuming the first two principal components encode the "coordinates" of the node, we simply re-scale the principal components of $h_i$ to match that of $\bar{h}_j$.

We view this approach as rather rudimentary. Namely, there are likely more informative vectors that encode richer information, such as information about neighboring nodes. However, we do find that the first two principal components have *some* causal role in the model's predictions.

We test our re-scaling intervention on 1,000 randomly generated contexts. For each context, assuming our underlying graph has $n$ nodes, we test "moving" the activations of the last token $i$ to all $n-1$ other locations in the graph. We then report the averaged metric across the resulting $1,000 \times n-1$ testcases.

We report 3 metrics: accuracy (Hit@1), Hit@3, and "accumulated probability mass" on valid tokens. Hit@1 (and Hit@3) report the percentage of times at which the top 1 (top 3) predicted token is a valid neighbor of the target node $j$. For "accumulated probability mass", we simply sum up the probability mass allocated to all neighbors (i.e., valid predictions) of the target node $j$.

Table 1 reports our results for our ring and grid tasks. We include results for re-scaling with 2 or 3 principal components, as well as null interventions and interventions with a random vector. Overall, we find that the principal components have *some* causal effect on the model's output predictions, but does not provide a full explanation.

|  | Ring | | | Grid | | | Hex | | |
|---|---|---|---|---|---|---|---|---|---|
|  | Hit@1 | Hit@3 | Prob | Hit@1 | Hit@3 | Prob | Hit@1 | Hit@3 | Prob |
| Interv. (n=2) | 0.61 | 0.91 | 0.6 | 0.57 | 0.95 | 0.55 | 0.30 | 0.32 | 0.69 |
| Interv. (n=3) | 0.77 | 0.96 | 0.76 | 0.68 | 0.98 | 0.65 | 0.42 | 0.46 | 0.82 |
| Null Interv. | 0.20 | 0.50 | 0.20 | 0.17 | 0.33 | 0.16 | 0.07 | 0.20 | 0.05 |
| Random Interv. | 0.17 | 0.47 | 0.19 | 0.16 | 0.37 | 0.17 | 0.06 | 0.18 | 0.05 |

Table 1: Intervention results for our ring and grid tasks. We demonstrate that often times, simply re-scaling the principal component for each token representation can "move" the token to a different position in the graph. However, we note that our simple re-scaling approach does not perfectly capture a causal relationship between principal components and model predictions.

## C.5 EMPIRICAL SIMILARITY OF PRINCIPAL COMPONENTS AND SPECTRAL EMBEDDINGS

Theorem 5.1 predicts that if the model representations are minimizing the Dirichlet energy, the first two principal components will be equivalent to the spectral embeddings $(z^{(2)}, z^{(3)})$.

Here we empirically measure whether the first two principal components are indeed equivalent to the spectral embeddings. In Table 2, we measure the cosine similarity scores between the principal components and spectral embeddings.

## C.6 ACCURACY OF IN-CONTEXT TASKS WITH A CONFLICTING SEMANTIC PRIOR

What would happen when an in-context task which contradicts a semantic prior is given to a model? Namely, Engels et al. (2024) show that words like days of the week have a circular representation.

|  | $\lvert \cos(\text{PC } 1, \boldsymbol{z}^{(2)}) \rvert$ | $\lvert \cos(\text{PC2}, \boldsymbol{z}^{(3)}) \rvert$ |
| --- | --- | --- |
| Grid | 0.950 | 0.954 |
| Ring | 0.942 | 0.930 |
| Hex | 0.745 | 0.755 |

Table 2: Absolute value of cosine distances of principal components from model activations and spectral embeddings. We empirically observe that in practice, these coordinates end up being very similar. For the grid and hexagon, we use principal components from the last layer, while for the ring, we use an earlier layer (layer 10) in which the ring is observed.

In our experiment, we randomly shuffle tokens for days of the week (i.e., tokens {Mon, Tue, Wed, Thu, Fri, Sat, Sun} to define a new ring, and give random neighboring pairs from the newly defined ring as our in-context task.

Figure 14 demonstrates the accuracy when given an in-context task that is contradictory to a semantic prior. Interestingly, we first observe the model make predictions that reflects the original semantic prior (pink). This accuracy drops very quickly as the model captures that the semantic rule is not being followed. With more exemplars, we see a slow decay of the remaining semantic accuracy and a transition in the model's behavior as it begins to make predictions that reflect the newly defined ordering of our ring (blue).

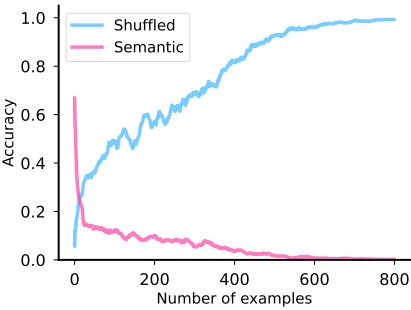

Figure 14: **In-context structure overrides semantic prior.** Given an in-context task that contradicts a model's semantic prior, we observe the model transition from making predictions that adhere to the semantic prior (pink) to predictions that reflect the newly defined in-context task.

Furthermore, in Fig. 15, we quantify the Dirichlet energy computed only from certain PC dimensions. We find that energy minimization happens in the dimensions corresponding to the in-context structure.

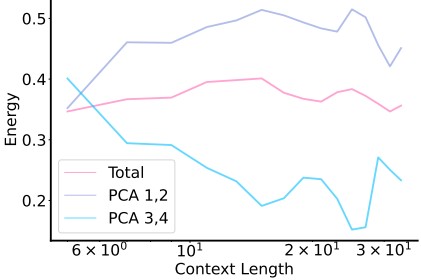

Figure 15: **Energy minimization happens in the in-context component dimensions.** We show the Dirichlet energy depending on the context given when taking 1) all 2) semantic (PCA 1,2) 3) in-context (PCA 3,4) dimensions. We show that energy minimization happens in PCA 3,4 corresponding to the in-context dimensions.

## C.7 Additional Empirical Verifications of Transition Predictions

Here we provide additional details for empirically verifying our predictions for model transitions.

Figures 16, 17, and 18 demonstrate detailed accuracy curves for a wide range of graph sizes.

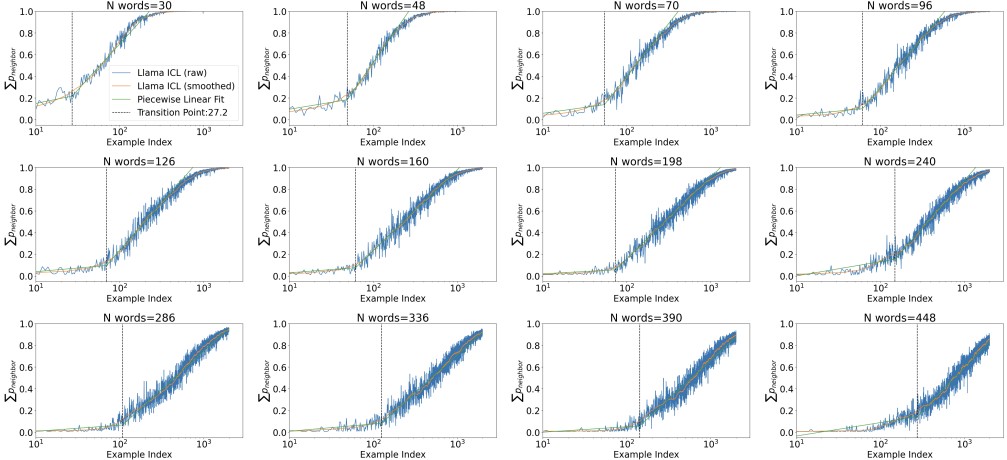

Figure 16: **Emergent behavior for varying task complexity (graph size) for the Hexagonal task.** We plot the accuracy for varying levels of complexity (graph size) for the hexagonal in-context task. Interestingly, regardless of graph size, we see an abrupt, discontinuous change in the model's performance. Figure 19 demonstrates that we can predict when such abrupt change can be expected as a function of task complexity.

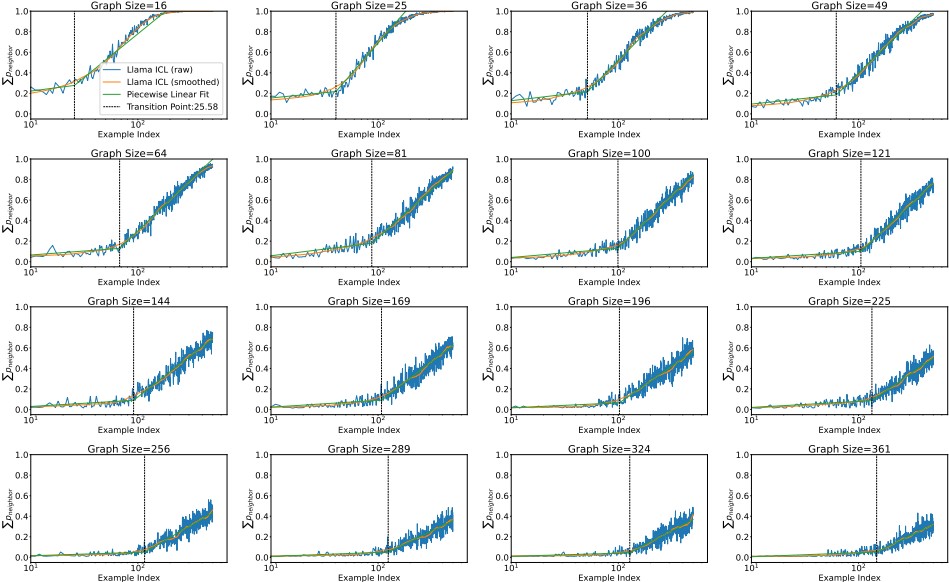

Figure 17: **Emergent behavior for varying task complexity (graph size) for the grid task.** We plot the accuracy for varying levels of complexity (graph size) for the grid in-context task. Interestingly, regardless of graph size, we see an abrupt, discontinuous change in the model's performance. Figure 8 demonstrates that we can predict when such abrupt changes can be expected as a function of task complexity.

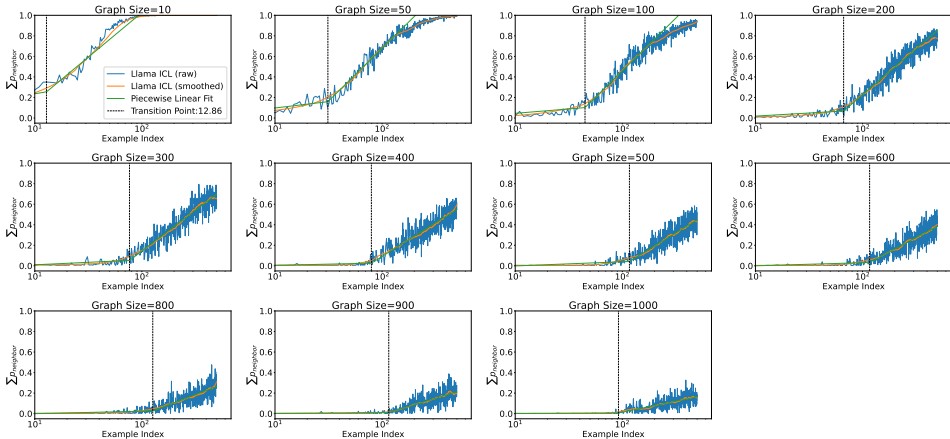

Figure 18: **Emergent behavior for varying task complexity (graph size) for the ring task.** We plot the accuracy for varying levels of complexity (graph size) for the ring in-context task. Interestingly, regardless of graph size, we again see an abrupt, discontinuous change in the model's performance.

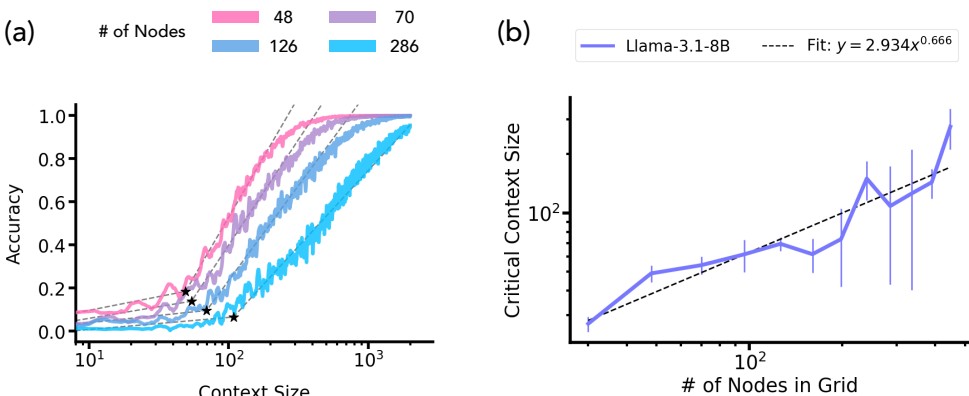

Figure 19: **In-context emergence in a Hexagonal graph tracing task.** We analyze the in-context accuracy curves as a function of context-size inputted to the model. The graph used in this experiment is an $m \times m$ grid, with a varying value for $m$. (a) The rule following accuracy of a graph tracing task. The accuracy show a two phase ascent. We fit a piecewise linear function to the observed ascent to extract the transition point, which moves rightwards with increasing graph size. (b) Interestingly, the transition point scales as a power-law in $m$, i.e., the number of nodes in the graph.

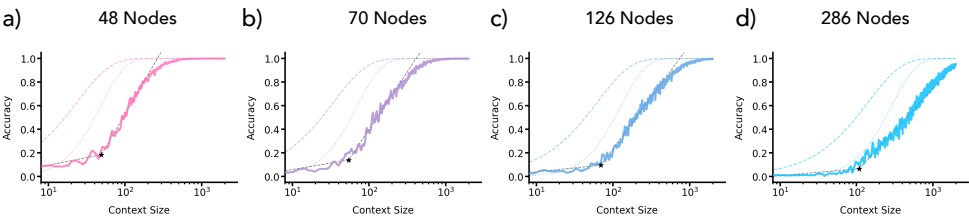

Figure 20: **Hexagonal graph tracing accuracies compared to the memorization solution** The rule following accuracies on the hexagonal graph compared to the memorization model in Sec. 4.1. Hexagonal graph with a) 48 b) 70 c) 126 d) 286 nodes. Generally we find that the hexagonal graph tracking accuracy from Llama-3.1-8B (Dubey et al., 2024) is lower than the 1,2-shot memorization model, indicating that there might be a different underlying process.

## D    LIMITATIONS

We emphasize that our work has a few limitations. Namely, PCA, or more broadly, low dimensional visualizations of high dimensional data can be difficult to interpret or sometimes even misleading. Despite such difficulties, we provide theoretical connections between energy minimization and principal components to provide a compelling explanation for why structures elicited via PCA faithfully represent the in-context graph structure. Second, we find a strong, but nevertheless incomplete, causal relationship between the representations found by PCA and the model's predictions. We view the exact understanding of how these representations form, and the exact relationship between the representations and model predictions as an interesting future direction, especially given that such underlying mechanism seems to depend on the scale of the context.

