# OpenReview forum: "ICLR: In-Context Learning of Representations"
_ICLR.cc/2025/Conference — ICLR 2025 Poster_

### Official Review · Reviewer_hdrb · 2024-11-03

**Soundness:** 3
**Presentation:** 3
**Contribution:** 2
**Rating:** 6
**Confidence:** 4

**Summary:**

This paper explores how in-context learning in large language models allows a reorganization of representations to fit novel task-specific semantics. The authors propose to override the pre-trained meaning of the words, thus the internal organization of the LLM representations with a semantic extracted from the neighbor relationships of names placed at the nodes of a graph.

The authors show that the LLM can internalize the graph structure learned in-context for sufficiently long input sequence sampled from the graph. They also connect their findings to energy minimization in graph and percolation theories, offering insight into how models might adapt internal representations based on in-context structure.

**Strengths:**

1. The paper presents a new task framework to study in-context reorganization in LLMs. It connects empirical observations with a theoretical foundation, connecting context-scaling with energy minimization.

2. The graph tracing task using well-defined structures (grid, ring) is an effective method for observing changes in LLM representations in a controlled way.

3. The paper is well-written and relatively easy to follow; the main goal is clearly stated, and the set of experiments are spot-on

**Weaknesses:**

The findings are based only on the analysis of llama3 representations, so extending the experiment to at least another model class would be easy.

The constructed task is somewhat artificial, and it is not immediately clear how we can extend these findings in the domain of natural language.

**Questions:**

Have you tried to see if your finding holds also on models other than Llama?

l371, 372 Have you tried to check empirically if the coordinates z2 and z3 are aligned with the principal axis you find with PCA?

What does DGP stand for in line 249?

---

> ### Author Response · Authors · 2024-11-22
> **Rebuttals**
>
> We thank the reviewer for their thoughtful feedback and positive remarks! We are delighted that the reviewer appreciated our novel task framework and found our theoretical foundation on spectral graph theory well suited to describe the experiments. The reviewer also praised that our experiments are an “ effective method for observing changes in LLM representations in a controlled way” and the paper is “well-written and relatively easy to follow”.
>
> Below, we respond to specific comments from the reviewer.
>
> ---
>
> > **The findings are based only on the analysis of llama3 representations, so extending the experiment to at least another model class would be easy.**
>
> We completely agree that verifying our claims on different models is essential to ensure their robustness! To this end, we have now re-run most experiments on Llama3.2-1B, Gemma2-2B, Gemma2-9B, and Llama3.1-8B-Instruct. *Results are shown in Appendix C.2 and we find identical conclusions across all settings!*
>
> ---
>
> > **The constructed task is somewhat artificial, and it is not immediately clear how we can extend these findings in the domain of natural language.**
>
> We note that our task is highly unnatural and artificial *by design*. The goal of our work is to identify whether LLMs can adapt their representations in a task specific manner, overriding the natural semantic organization they may have acquired during pretraining. Recent findings and trends in LLM research motivated our work:
>
> - [1a] points out that LLMs can learn a totally new language from in-context exemplars. While this study *suggests* that LLMs can learn new meanings, rules and relations from the input context, we aimed to study whether a task specific representation emerges. Most current work analyzing LLM representations focuses on how representations are organized after pre-training [1b,1c,1d]. As inference scaling and long context inference methods become more popular, we argue a research agenda focused solely on understanding how (if at all) representations develop in-context is becoming timely. We believe our work is a solid first step on this front.
>
> - There has been an enormous effort in the RL community to use LLMs for solving open-ended interactive tasks using Transformers or pretrained LLMs (often called in-context reinforcement learning; e.g., see [2a]). Such work adds policy rollouts or exploration trajectories into the model’s context, hoping that it yields a “world model” that the network can then exploit for performing goal-directed tasks. Our proposed graph tracing tasks were directly motivated by this literature, and we believe our work provides an empirical grounding to these in-context RL efforts: Our results show there is in fact a “world model” emergent by merely adding exploration trajectories (random walks on the graph)!
>
> Beyond the motivations listed above, we also highlight that a fascinating series of papers has demonstrated humans’ ability to perform similar re-mapping of semantic concepts as we demonstrated in our work [3a, 3b]. These papers use a visual counterpart of exactly the same task as we proposed in our work, speaking to the scientific importance of our analysis!
>
> [1a] https://arxiv.org/abs/2403.05530
>
> [1b] https://arxiv.org/abs/2405.14860
>
> [1c] https://arxiv.org/abs/2311.03658
>
> [1d] https://arxiv.org/abs/2310.06824
>
> [2a] https://arxiv.org/abs/2306.14892
>
> [3a] https://elifesciences.org/articles/17086
>
> [3b] https://www.sciencedirect.com/science/article/pii/S009286742031388X
>
> ---
> ---
>
> **Further Questions.**
>
> > **Have you tried to see if your finding holds also on models other than Llama?**
>
> As mentioned above, we have now tested our findings on other models!
>
>
> ---
> > **l371, 372 Have you tried to check empirically if the coordinates z2 and z3 are aligned with the principal axis you find with PCA?**
>
> Great question! Indeed, we find that there is very high alignment – we observe the following cosine similarity scores between the principal components in which the graph shows up and z2, z3 from Theorem 5.1:
>
> |            |  cos(PC, z2)   |   cos(PC2, z3) |
> |----------|:---------------:|-----------------:|
> | Grid      |  0.95              |         0.954      |
> | Ring     |    0.94          |         0.93       |
> | Hex      | 0.745             |         0.755       |
>
> ---
> > What does DGP stand for in line 249?
>
> DGP stands for Data Generating Process. We did not realize this was the first place the acronym was being used. We have fixed this now; thank you for pointing this out!
>
> ---
> ---
>
> **Summary.** We thank the reviewer for their feedback that has helped us improve the robustness of our claims. We have now re-run the bulk of our experiments on several models and confirmed our findings generalize to them! We have also added experiments confirming the validity of our theoretical claims. In case our responses have addressed the reviewer’s concerns, we would be grateful if they can consider raising their score to support the acceptance of our work!

---

> > ### Author Response · Authors · 2024-11-24
> >
> > Dear Reviewer,
> >
> > We thank you again for your detailed feedback on our work. Given the discussion period ends soon, we wanted to check in if our provided responses address your concerns, and see if there are any further questions that we can help address.
> >
> > Thanks!

---

> ### Author Response · Authors · 2024-11-30
> **Summary of updates before discussion deadline**
>
> Dear Reviewer hdrb,
>
> Thank you for your thoughtful feedback that helped strengthen our work substantially. Given that the extended discussion period ends in two days, we wanted to check if you had the chance to review our responses and the extensive updates we've made to address your concerns.
> We recognize that we have made numerous changes which might take time to process, so we'd like to briefly highlight the key updates that directly address your main concerns:
>
> 1. **Addressing your concern about robust verification:** As you suggested, we've validated all our key findings **across multiple models (Llama3.2-1B, Gemma2-2B, Gemma2-9B, and Llama3.1-8B-Instruct) and graph structures**. These comprehensive results are presented in **new Figures 11-13 in Appendix C.2 and C.3**, demonstrating that our conclusions hold robustly across all settings.
>
> 2. **Addressing your concern about motivation for studying our task:** We've **substantiality revised our manuscript and strengthened the motivation** for our synthetic task by connecting it to recent work on in-context RL ([2a]) and human studies ([3a, 3b]) that demonstrate similar semantic remapping capabilities.
>
> 3. **Addressing your concern about theoretical validation:** We've quantified the alignment between principal components and theoretical predictions with **new experiments finding high cosine similarities**, fully addressing your concern.
>
> If you find these updates satisfactory, we would be grateful if you could consider raising your score to support the acceptance of our work. Of course, if you have any remaining questions or concerns, we are here to address them promptly.
>
> Thank you again for your time and valuable feedback!

---

> > ### Comment · Reviewer_hdrb · 2024-12-02
> >
> > I apologize for the late reply.
> > Thank you for the work you have done in the rebuttal and the updates in the manuscript. I think this work can be of interest to the user community. Therefore, I have raised my score to 6.

---

### Official Review · Reviewer_EbvH · 2024-11-03

**Soundness:** 2
**Presentation:** 2
**Contribution:** 3
**Rating:** 6
**Confidence:** 4

**Summary:**

The paper studies how the different representations are organized in a large language model’s internals. It reveals the sudden reorganization of representations as the context length increases.

**Strengths:**

1. The problem is interesting and important.
2. The experiment results are interesting.

**Weaknesses:**

1. Lack of quantitative metrics of “the representations mirror the grid/ring structure” in Section 2: It is unclear how the PCA plot relates to the conclusions that the representations reflect the graph structure. The results in Section 4 may be helpful but require more discussion.
2. The use of PCA: The PCA is useful for visualization but is not convincing enough to draw rigorous conclusions. Without the manually added vertices in Figures 1(c), 2(c), and 3(c), the results are not interpretable.
3. Equ. (4) is the probability of seeing one node with each token randomly sampled with replacement. It does not match the data generating procedure with the first token randomly sampled, and the remaining tokens are generated through random walk.
4. I do not find the percolation theory fits here. Although Figure 8(b) indicates a strong result, it is unclear how y and x connect to the percolation theory.

**Questions:**

1. What is the layer of the figure 1(c)
2. In Section 3.1, it is claimed that a window of $N_w=200$ preceding tokens is included in the computation of representations. This number is greater than the minimal context length of Figure 2(b), which is 100.
3. How does Figure 3(c) prove that the LLM learns the in-context non-semantic structure?
4. The percolation relates to the thresholds of having an infinite connected path. Can authors explain more explicitly about its relation to the problem considered in the paper?

---

> ### Author Response · Authors · 2024-11-22
> **Rebuttals (1/4)**
>
> We thank the reviewer for their detailed feedback! We are pleased that the reviewer found the problem we tackled and the experiments we conducted interesting and important.
>
> Below, we respond to specific comments from the reviewer. In summary, the reviewer is concerned that 1) we make our claims based on PCA visualizations and 2) that there are some clarifications to make on our design choices. We address these concerns by 1) clarifying that the findings themselves are drawn from quantitative metrics, such as Dirichlet energy with PCA used solely for demonstrating the results visually and motivating more rigorous validation in the following section, and 2) revising the manuscript to better introduce percolation theory.
>
> ----
> ----
> > **Lack of quantitative metrics of “the representations mirror the grid/ring structure” in Section 2: It is unclear how the PCA plot relates to the conclusions that the representations reflect the graph structure. The results in Section 4 may be helpful but require more discussion.**
> > **The use of PCA: The PCA is useful for visualization but is not convincing enough to draw rigorous conclusions. Without the manually added vertices in Figures 1(c), 2(c), and 3(c), the results are not interpretable.**
>
> We respond to the two comments above simultaneously since they make the same broader argument, i.e., that PCA by itself is an insufficient technique to make a conclusive statement on whether the model is forming representations corresponding to the in-context graph.
>
> First, we note that we completely agree with the reviewer that PCA can be insufficient to analyze representations and make conclusive statements on structures therein. However, this is precisely the gap that Section 4 and Section 5 of the paper help us cover in both an empirical and theoretical manner. Specifically, these sections help us avoid PCA’s usual pitfalls in the following manner.
>
> - **Dirichlet energy as the quantitative metric of structure.** In Section 4, we provide a precise quantitative analysis of Dirichlet energy, which measures how aligned the representations are with respect to some ground truth graph structure (we have in fact expanded this analysis to several model families now). Achieving the minimum of this metric requires for the graph adjacency matrix to be recoverable from the model representations! To fully address your feedback, we have now conducted both PCA (Fig. 11) and Dirichlet curve (Fig. 12) experiments, each consisting of 12 distinct setups, including 4 different models (Llama3.2 1B, 8B-Instruct and Gemma2 2B/9B) and 3 types of graph structures (square grid, ring, and hexagonal grid)! (New experiments added during rebuttals; see Appendix C.2). Given this substantial quantitative validation, we argue the standard pitfalls of low-dimension visualization methods like PCA are unlikely to confound our results.
>
> - **Section 5: Proving that PCA visualizations are faithful.** Building on the above, in Section 5, we theoretically characterize why PCA results ended up demonstrating the graph structure: assuming Dirichlet energy is minimized (which we empirically demonstrated to be the case as in the above), we prove that top-2 PCA components will in fact be equal to spectral embeddings of the in-context graph (see the new experiment discussed below for further empirical evidence for this claim). That is, we can prove that the PCA visualizations represent the structure of the graph *because they ought to!*
>
>
>     - **New experiments confirm the theoretical motivation for using PCA.** In Theorems 5.1, B.1, we prove that with Dirichlet energy minimization, we should expect to see our graphical structures emerge within the early principal components (see Fig 6, 7). To further validate this claim and demonstrate the PCA visualizations we report in the paper are faithful, we now report the cosine similarity of the spectral embeddings of our ground-truth graph, which form the basis of Theorem 5.1 (denoted z2, z3), and the principal axes of our representations (denoted PC1, PC2). Results are shown in the table below and we see that for all graph topologies, top principal components have high similarity with the ground-truth graphs’ spectral embeddings! This implies the principal components are in fact *the basis* for visualizing the representations of our model.
>
> |            |  cos(PC, z2)   |   cos(PC2, z3) |
> |----------|:---------------:|-----------------:|
> | Grid      |  0.95              |         0.954      |
> | Ring     |   0.94          |         0.93       |
> | Hex      | 0.745             |         0.755       |
>
> **[Continued below...]**

---

> ### Author Response · Authors · 2024-11-22
> **Rebuttals (2/4)**
>
> - **Further validation: The PCA dimensions are causally relevant.** Finally, to further validate the faithfulness of PCA visualizations, we have added another experiment that demonstrates the top principal components encode information that is causally relevant to how our model solves the task! Specifically, in App. C.4, we perform the following intervention on the representation of the last token shown to the model in context: (i) remove the representation’s component along the top two PCA dimensions, and (ii) add another concept’s representation’s components along the top two PCA dimensions. This should ideally make the model believe that it is at a different node in the graph, i.e., not the node indicated by the last token shown in context. Our results show this modifies the model’s prediction for the next token in a largely predictable manner! This indicates that the principal components not only represent the graphical structures, but that they also capture causally relevant information that is used by the model to solve our in-context learning tasks.
>
> Overall, we emphasize that pitfalls of PCA do not affect our claims, since we perform an extensive quantitative analysis to corroborate our claims and theoretically demonstrate the faithfulness of PCA as a visualization technique for our work.
>
> *We also note that we have updated the paper in Section 3 to better reflect these arguments and highlight that a solid quantitative analysis is available in Section 4.*
>
> ----
>
> > **Equ. (4) is the probability of seeing one node with each token randomly sampled with replacement. It does not match the data generating procedure with the first token randomly sampled, and the remaining tokens are generated through random walk.**
>
> Please note that we analyze two variants of our graph tracing task, as noted on Lines 143–145 (“we sample a random walk from a pre-defined graph and then give either the entire walk or parts of it (“traces”) to the model, expecting it to understand the overall structure of the graph”).
>
> That is, in the first variant of our task, we perform a random walk and simply feed to walk into the model (as shown in Figure 1); in the second variant, we show the model only parts of the walk at a given time (as shown in Figure 2). This latter variant thus allows random sampling with replacement. We note we have clarified this distinction better in the updated manuscript now, and apologize if this was unclear in the original draft!
>
> ---
>
> **[Continued below...]**

---

> ### Author Response · Authors · 2024-11-22
> **Rebuttals (3/4)**
>
> > **I do not find the percolation theory fits here. Although Figure 8(b) indicates a strong result, it is unclear how y and x connect to the percolation theory.**
>
> Thank you for your question regarding the role of percolation theory in our analysis. First, we would like to clarify that we currently consider the percolation model as merely a “candidate hypothesis” (as noted in the title of Sec. 5.1) for explaining the novel “transition” observed in the accuracy curve as we scale in-context samples. While we provide substantial supportive evidence for our hypothesis (e.g., Fig. 8b), we do not yet regard this as formal proof. We appreciate your feedback and are happy to further clarify this point in the writing if needed.
>
> Regarding our rationale for considering the percolation hypothesis, we refer the reviewer to Section 5.1, where we discuss the relation between percolation and the in-context task in Section 5 (Connectivity and Energy Minimization). Our primary argument can be summarized as follows: “We expect an emergent structure where the model connects the whole graph into a single connected component.” Specifically, we claim once the model has seen enough edges on the graph, it will be able to transition from trying to find semantic relations between pairs (or simply memorizing the examples) to an inference time graph neighbor search task. We can also formulate this using the Bayesian picture of In-Context Learning (Xie et al 2021). Given a small amount of examples, the best hypothesis explaining the data might be some semantic relation. However when a large number of examples are given, the simplest hypothesis explaining the data becomes the one where there is a limited underlying graph which produces the data. Once the model has received enough examples to construct this graph, we expect 1) an emergent representation of the whole graph and 2) a discontinuity in accuracy with the shift of hypothesis. This discontinuity of performance is shown in Figure 5 and 8. In Fig 8.b, we show that the point of discontinuity, i.e. the percolation threshold, scales similarly to n^0.5 as expected from percolation theory. This result cannot be explained by a simple solution like a fixed fraction of the graph being observed since (#of edges observed)/(# of total edges) is only expected to decrease linearly. In fact, we find that at n=25, the discontinuity happens when ~40 edges are seen while for n=144 it happens when ~90 edges are seen, suggesting that this discontinuity is not just happening at the graph size either. These observation suggests that there *might* be a graph percolation process underlying these accuracy curves.
>
> ----
> ----
> ----
> **Further Questions.**
>
> > **Q1. What is the layer of the figure 1(c)**
>
> These visuals are from the residual stream following layer 26 – we have clarified this in Fig. 1’s caption now. Thank you.
>
> ----
> > **Q2. In Section 3.1, it is claimed that a window of Nw=200 preceding tokens is included in the computation of representations. This number is greater than the minimal context length of Figure 2(b), which is 100.**
>
> Thank you for raising this point and we apologize for the confusion! We mistakenly forgot to qualify that when the context size (denoted $N_c$) is smaller than the window size used for computing representations (denoted $N_w$), we simply use the entire context as the window. That is, we set $N_w = N_c$. We have now clarified this in the manuscript: “*At each timestep, we look at a window of Nw (=50) preceding tokens (or all tokens if the context length is smaller than Nw)*.”
>
> ----
>
> > **Q3. How does Figure 3(c) prove that the LLM learns the in-context non-semantic structure?**
>
> Please note that the novel semantics in this experiment are encoded by a new graph which, itself, has a ring topology as well. Specifically, the ring goes as “Tuesday -> Saturday -> Wednesday -> Sunday -> Thursday -> Monday -> Friday -> Tuesday”. Now, if we were to take the original graph represented by days-of-the-week (which itself forms a ring, as shown via pink edges in Figures 3a, 3b) and add to it edges corresponding to this new graph (as shown via blue edges in Figure 3a, 3b), we would retrieve a star structure. However, since our random graph tracing task relies on random walk corresponding to the novel graph, as we expect, the model representations for days-of-the-week get arranged according to the novel ring structure, as shown in Figure 3(c).
>
> We also note that following reviewer’s suggestions before, to corroborate these results in a more quantitative manner, we have now added results corresponding to the Dirichlet energy minimization experiment for this setup. These results, as reported in Fig. 15 in the appendix, corroborate in a quantitatively precise manner that the representations in the 3rd and 4th principal components are getting arranged according to the novel ring structure.
>
> ----
>
> **[Continued below...]**

---

> > ### Author Response · Authors · 2024-11-22
> > **Rebuttals (4/4)**
> >
> > ---
> > > **Q4. The percolation relates to the thresholds of having an infinite connected path. Can authors explain more explicitly about its relation to the problem considered in the paper?**
> >
> > Please see our comment in response to the fit of percolation theory for our results above.
> >
> > ---
> > ---
> > **Summary.** We thank the reviewer for their detailed feedback, which has helped us better contextualize our PCA analysis and emphasize in the paper why these visualizations are faithful to our main claim, i.e., LLMs can more in-context morph semantics of a concept. We hope our responses and new experiments have helped address the reviewer’s concerns, and, if so, we hope they will consider increasing their score to support the acceptance of our work.

---

> > > ### Author Response · Authors · 2024-11-24
> > >
> > > Dear Reviewer,
> > >
> > > We thank you again for your detailed feedback on our work. Given the discussion period ends soon, we wanted to check in if our provided responses address your concerns, and see if there are any further questions that we can help address.
> > >
> > > Thanks!

---

> ### Author Response · Authors · 2024-11-30
> **Summary of updates before discussion deadline**
>
> Dear Reviewer EbvH,
>
> We want to especially thank you for your detailed feedback that highlighted critical areas for improvement in our work. Your specific concerns about quantitative metrics and PCA validation motivated us to conduct extensive new experiments. As the discussion period ends in two days, we would like to summarize how we've addressed each of your key points:
>
> 1. **Addressing your concern about "lack of quantitative metrics" and PCA validation:**
>    * Following your feedback, we conducted comprehensive experiments across four additional models (Llama3.2-1B, Gemma2-2B, Gemma2-9B, and Llama3.1-8B-Instruct) and three distinct graph structures (square grid, ring, and hexagonal grid), with results in Figs. 11, 12, and 13 (App. C.2 and C.3)
>    * To address your point about PCA interpretation, we quantified cosine similarity between graph spectral embeddings and principal components (Table 2, App. C.5), demonstrating high alignment (e.g., cos(PC1,z2)=0.95 for grid topology)
>    * Responding to your concern about rigorous conclusions, we demonstrated causal relevance of PCA dimensions through intervention experiments (App. C.4)
>
> 2. **Regarding your questions about semantic prior analysis:**
>    * Based on your question about Fig. 3(c)'s interpretation, we strengthened our analysis with quantitative results in Fig. 15, measuring energy in both semantic structure (first/second PCs) and in-context structure (third/fourth PCs)
>    * We demonstrated the expected energy decrease in in-context structure dimensions with increasing context
>
> 3. **Clarifying methodology per your questions:**
>    * Updated window size computation explanation in Section 3.1, directly addressing your question about Nw
>    * Clarified the distinction between random walk and random sampling variants you noted
>    * Strengthened theoretical motivation for PCA in Section 3, connecting to spectral graph theory
>
> Importantly, all these new experiments demonstrate that our findings hold robustly across different model architectures and graph structures, addressing your fundamental concern about the soundness of our conclusions.
>
> Given your crucial role in strengthening our work through these validations, we would greatly appreciate it if you could reconsider assessment of our work in light of the new updates. If you have any remaining questions or concerns, we are committed to addressing them immediately before the discussion period ends.

---

> > ### Comment · Reviewer_EbvH · 2024-11-30
> >
> > Thank the authors for their response. I appreciate the detailed explanations for using PCA, and I would like to increase my score.
> >
> > However, I hope the authors seriously revise the percolation part. The current argument is still far from convincing. The experiments are already interesting enough. Adding the percolation part only deteriorates the results.

---

> ### Author Response · Authors · 2024-11-30
>
> Thank you for the prompt response! We promise to address your concern about the percolation model in the final version of the paper. To emphasize, the section on percolation is only half-a-page, and before that the concept is mentioned only twice (abstract and intro). **Editing these parts is thus very easy!** Specifically, we will make the following changes to the manuscript.
>
> - Remove any phrases in abstract and introduction that suggest percolation is a mechanistic explanation for emergent representation formation
>
> - Keep the transition point scaling results to suggest there is likely an interesting phenomenon at play here, but remove the discussion about percolation transition
>
> - Add 2--3 lines in Conclusion / Discussion that argue for the need of a theoretical model to explain the transition point, pitching percolation as a possible (but certainly not confirmed) candidate.
>
> We hope these edits will help address the reviewer's concerns!

---

### Official Review · Reviewer_a5tz · 2024-11-03

**Soundness:** 3
**Presentation:** 4
**Contribution:** 3
**Rating:** 6
**Confidence:** 4

**Summary:**

This work designs a synthetic task to determine to what extent LLM  representations reflect in-context semantics rather than those observed during pre-training.

To do this they generate a graph structure, where nodes consist of tokens that are highly likely to be observed during pre-training, but with novel dependencies defined by the structure. They then generate sequences following a random walk across the graph, which are fed into the LLM 'in-context' (i.e. without weight updates). They then show that the LLM can learn to organise its representations such that they mimic the underlying graph structure.

The task is to some extent adversarial, because the context neighbourhoods defined by the graph differ from the standard semantics of the tokens.

Evidence presented is both qualitative (using PCA) and quantitative (various measures for how well representations conform to the graph structure).

**Strengths:**

The paper is excellently written, and the figures clearly get the central findings across the reader, making it a pleasure to read. The experimental design is original, elegant and well crafted in order to probe the hypothesis. Finally, the findings provide an interesting insight in the abilities of LLMs to zeroshot generalise to novel connective structures.

**Weaknesses:**

The work relies quite heavily on PCA, which the authors admit can be somewhat misleading, though they provide theoretical justifications for why this reliance is valid.

On a higher level I am not quite sure whether the finding is particularly surprising. Natural language is supposed to have similar causal relationships modelled by dependency parses, and LLMs are capable of modelling it successfully. Moreover, they are supposedly able to learn to translate hitherto unseen languages entirely in context ([1]), which constitutes a much more complex graph than the ones assessed in this paper. As a result I am not certain that the findings expose a new capability. However, as stated in the strengths section the paper provides a very nice framework for understanding this capability which should bias towards acceptance.

[1] https://arxiv.org/abs/2403.05530

**Questions:**

Have you or do you plan to extend your investigation to more complicated structures? The ones presented in the paper are fairly simple and it would be interesting if there are certain forms that are harder to represent.

---

> ### Author Response · Authors · 2024-11-22
> **Rebuttals (1/2)**
>
> We thank the reviewer for the feedback and positive review! We are especially glad that the reviewer found our paper “excellently written”,  “pleasure to read”, and found the experiments “original, elegant and well crafted”. The reviewer accurately grasped our main findings about how a model organizes representations depending on in-context semantics, and we appreciated their high scores on contribution, soundness, and presentation of our work!
>
> Below, we respond to specific comments from the reviewer.
>
> ---
> ---
>
> > **The work relies quite heavily on PCA, which the authors admit can be somewhat misleading, though they provide theoretical justifications for why this reliance is valid.**
>
> This is a fair point! However, we believe standardly known pitfalls of PCA are unlikely to affect our results, as summarized below (we have added these comments to the paper as well; see Section 7, Discussion).
>
> - **New experiments ground the theoretical motivation for using PCA.** As noted by the reviewer, our use of PCA for visualizations is motivated by the fact that our theorems (5.1, B.1) predict that with Dirichlet energy minimization, we should expect to see our graphical structures emerge within the early principal components (see Fig 6, 7). To further emphasize that this motivation is well-grounded, *we have now run new experiments to validate the theorem more precisely*. Specifically, we report the cosine similarity of the spectral embeddings of our ground-truth graph, which form the basis of Theorem 5.1 (denoted z2, z3), and the principal axes of our representations (denoted PC1, PC2). Results are shown in the table below and we see that for all graph topologies, top principal components have high similarity with the ground-truth graphs’ spectral embeddings! This implies the principal components are in fact *the basis* for visualizing the representations of our model.
>
> |            |  cos(PC, z2)   |   cos(PC2, z3) |
> |----------|:---------------:|-----------------:|
> | Grid      |  0.95              |         0.954      |
> | Ring     |    0.94          |         0.93       |
> | Hex      | 0.745             |         0.755       |
>
>
> - **The PCA dimensions are causally relevant.** To further ground the validity of PCA as the right method for our analysis and to demonstrate that its use does not confound our results, we have now run experiments that top principal components retrieve information that is causally relevant to how our model solves the task! Specifically, in App. C.4, we perform the following intervention on the representation of the last token shown to the model in context: (i) remove the representation’s component along the top two PCA dimensions, and (ii) add another concept’s representation’s components along the top two PCA dimensions. This should ideally make the model believe that it is at a different node in the graph, i.e., not the node indicated by the last token shown in context. Our results show this modifies the model’s prediction for the next token in a largely predictable manner! This indicates that the principal components not only represent the graphical structures, but that they also capture causally relevant information that is used by the model to solve our in-context learning tasks.
>
> - **Our findings do not rely on PCA visualizations themselves.** Beyond the well-grounded motivations noted above, we also note that most of our conclusions are not drawn from the PCA visualizations by themselves. Instead, we provide precise quantitative scores to ground our claims throughout the paper. For example, the emergence of a task specific representation is quantified by computing graph spectral energy in Fig. 4, and the ability of our model to solve the task is quantified via an accuracy measure. The robustness of our findings is also grounded by the fact that they generalize to several experimental settings, including different graph topologies, graph sizes (we scale up to 1000 nodes!), and now even different model families (new experiments added during rebuttals; see Appendix C.2). Given this substantial quantitative corroboration, we argue the standard pitfalls of low-dimension visualization methods like PCA are unlikely to confound our results.
>
> ---
>
> **[Continued below...]**

---

> ### Author Response · Authors · 2024-11-22
> **Rebuttals (2/2)**
>
> > **On a higher level I am not quite sure whether the finding is particularly surprising. Natural language is supposed to have similar causal relationships modelled by dependency parses, and LLMs are capable of modelling it successfully.... However, as stated in the strengths section the paper provides a very nice framework for understanding this capability which should bias towards acceptance.**
>
> While we agree that it might not be surprising that the model can perform such novel in-context tasks, to our knowledge, we are the first to demonstrate that a fundamental reorganization of concepts can occur in an entirely in-context manner. We believe this finding is crucial for several reasons, as summarized below.
>
> - Most current work analyzing model representations focuses on how representations are organized after pre-training [1a,1b,1c,1d]. As inference scaling and long context inference methods become more popular, we argue a research agenda focused solely on understanding how (if at all) representations develop in-context is becoming timely. We believe our work is a solid first step on this front.
>
> - There has been an enormous effort in the RL community to use LLMs for solving open-ended interactive tasks using Transformers or pretrained LLMs (often called in-context reinforcement learning; e.g., see [2a]). Such work adds policy rollouts or exploration trajectories into the model’s context, hoping that it yields a “world model” that the network can then exploit for performing goal-directed tasks. Our proposed graph tracing tasks were directly motivated by this literature, and we believe our work provides an empirical grounding to these in-context RL efforts: Our results show there is in fact a “world model” emergent by merely adding exploration trajectories (random walks on the graph)!
>
> - A fascinating series of papers has demonstrated humans’ ability to perform similar re-mapping of semantic concepts as we demonstrated in our work [3a, 3b, 3c, 3d]. In line with several recent papers using neural networks, including LLMs, for better understanding human cognition [3e], our results indicate we may now have an experimental system for studying the mechanistic basis of how humans re-organize novel concepts! We have added a discussion of this line of work to the paper as well.
>
> [1a] https://arxiv.org/abs/2405.14860
>
> [1b] https://arxiv.org/abs/2311.03658
>
> [1c] https://arxiv.org/abs/2310.06824
>
> [1d] https://arxiv.org/abs/1301.3781
>
> [2a] https://arxiv.org/abs/2306.14892
>
> [3a] https://elifesciences.org/articles/17086
>
> [3b] https://www.sciencedirect.com/science/article/pii/S009286742031388X
>
> [3c] https://elifesciences.org/reviewed-preprints/101134
>
> [3d] https://www.nature.com/articles/s41467-020-18254-6
>
> [3e] https://arxiv.org/abs/2410.20268
>
>
> ----
> ----
> **Further Questions.**
>
> > **Have you or do you plan to extend your investigation to more complicated structures? The ones presented in the paper are fairly simple and it would be interesting if there are certain forms that are harder to represent.**
>
> We agree that more complicated structures will be interesting to study. During our investigations, we did try another lattice structure, specifically triangular grids. Our results generalized for this structure as well, but we preferred doing our exhaustive batch of experiments on just the rectangular and hexagonal lattices since they are more complicated than triangular ones.
>
> Beyond such lattice structure, we think hierarchical structures (e.g., trees) will be interesting to study. We see probabilistic context-free grammars as a potential setup for studying such structures. However, we believe PCA is unlikely to enable visualizations of such tree structures. Finding alternative strategies seems currently out of scope for the scope of this work, but we intend to explore it in future work!
>
> ----
> ----
>
> **Summary.** We thank the reviewer for their detailed feedback that has helped us better contextualize our use of PCA as a visualization tool for our experiments! To this end, we have added new experiments that ground our theory on why PCA offers the right basis for visualization and that demonstrate the causal relevance of directions extracted via PCA. We have updated the discussion section and added references to these new experiments in the main paper. In case our responses have addressed the reviewer’s concerns, we would be grateful if they can consider championing our paper for acceptance!

---

> > ### Author Response · Authors · 2024-11-24
> >
> > Dear Reviewer,
> >
> > We thank you again for your detailed feedback on our work. Given the discussion period ends soon, we wanted to check in if our provided responses address your concerns, and see if there are any further questions that we can help address.
> >
> > Thanks!

---

> > > ### Comment · Reviewer_a5tz · 2024-12-03
> > >
> > > Thank you for your responses and additional work. I would strongly recommend that some of the context setting discussions above be incorporated into the final manuscript.

---

### Official Review · Reviewer_KJBF · 2024-11-07

**Soundness:** 3
**Presentation:** 4
**Contribution:** 2
**Rating:** 8
**Confidence:** 4

**Summary:**

The paper investigates how LLMs form representations for novel concepts during in-context learning. Authors design a synthetic graph navigation task where the model has to learn from random walks between graph nodes are referenced as words from the training set (e.g. “apple”), meaning that the model has to learn a new representation for these words to accurately predict the next node in the graph traversal. They refer to this problem as “in-context graph tracing”.

Authors then study the hidden representations formed by LLMs as they learn this task in-context, visualizing their first few principial components. They find that, after a certain number of examples, the model abruptly re-organizes the representation space in a way that facilitates the graph traversal task, as learned from the context. They also find cases where the in-context learned representations do not dominate  the first principal components, but they are still present in subsequent components (e.g. 3rd and 4th in Figure 3). Finally, authors demonstrate a connection between the process of in-context learning of node representations and a form of graph energy minimization.

**Strengths:**

- The paper investigates an interesting phenomena that may further our understanding of how LLMs learn in-context;
- The paper includes a simple and easy-to-reproduce experiment that can facilitate future analysis of LLM representations;
- The paper is generally well-written, if a bit unconventionally structured. It follows the natural flow of an investigation into how representations change during in-context learning;
- Authors go to reasonable length to ablate alternative hypotheses in their analysis.

**Weaknesses:**

- The paper focuses primarily on just a single model: Llama 3 8B. It is possible that some of the authors findings reflect not the general ability of LLMs, but an artifact of Llama 3 models, or a property of models of this particular size. The paper would arguably be improved if authors verify their findings on other models: both of different size (e.g. 1B / 70B / 405B), different family (gemma, qwen, mistral, opt, etc) and of different type (e.g. instruct vs non-instruct). Failing that, authors could at least clarify this as potential limitation.

- The part where authors argue that the in-context learning is linked to Dirichlet energy minimization is interesting, but (arguably) a bit contrived. As authors explain, energy minimization would be solved simply by assigning all nodes to the same embedding. Perhaps it would be best to further explore the analogy with graph embedding learning as a possible alternative? (e.g. distance-preserving embeddings or similar)

- While I don't count this as a significant weakness, the paper could arguably be improved by further exploring *what* in the model causes the abrupt shift in representations. (e.g. is there any connection with induction heads, transformer circuits, or any other mechanism that could be responsible for the change).

**Questions:**

### Questions:

Note that these questions are mostly asked out of curiosity and do not affect my overall recommendation.

1. Authors found that there is a shift in concept representations after the model has seen enough context. **Could this mean that the model does not 'understand' early examples in real few-shot learning tasks?** For instance, could it mean that the model would not be able to reason about information if it requires understanding the sequences it 'read' before the change in representations? If yes, would it mean that the model would benefit from repeating the early in-context examples again, once the model 'understands' them?

2. What happens to the representation shift if you increase the number of concepts? Is there a point by which the model is no longer able to shift the structure properly to match the in-context 'meanings'?

### Minor:

L186 task-specific? (more popular with a dash)

L269 “Given that we empirically do not observe this to be the case, we can safely assume this trivial solution does not arise in our experiments.” - slightly odd phrasing - you make an observation about your experiments that implies an assumption about the same experiments? Perhaps it would be best to paraphrase / clarify the footnote.

---

> ### Author Response · Authors · 2024-11-22
> **Rebuttals (1/2)**
>
> We thank the reviewer for their detailed feedback and positive review. We are glad they found our proposed experiments interesting, useful for understanding in-context learning in LLMs, and easy to reproduce! The reviewer also found our paper well-written and praised our control experiment conducted for rigorousness, and we appreciated their high scores on soundness and presentation of our work! Finally the reviewer accurately summarized our main findings: 1) LLMs develop task specific representations given in-context exemplars. 2) When there is a strong semantic representation, LLMs can use subsequent dimensions to represent the in-context representations. 3)The learning of task specific representations is connected to a graph energy minimization process.
>
> Below, we respond to specific comments by the reviewer.
>
> ---
> ---
>
> > **The paper focuses primarily on just a single model: Llama 3 8B. It is possible that some of the authors findings reflect not the general ability of LLMs, but an artifact of Llama 3 models, or a property of models of this particular size.**
>
> Thank you so much for your concrete suggestions to strengthen our claims! To fully reflect your feedback, we have now re-run most experiments on *Llama3.2-1B, Gemma2-2B, Gemma2-9B, and Llama3.1-8B-Instruct.* Results are shown in Appendix C.2 and we find identical conclusions across all settings!
>
> ---
>
> > **The part where authors argue that the in-context learning is linked to Dirichlet energy minimization is interesting, but (arguably) a bit contrived. As authors explain, energy minimization would be solved simply by assigning all nodes to the same embedding….**
>
> The possibility of a trivial solution, i.e., one that assigns the same embedding to all nodes, is indeed a possible confounder in Dirichlet energy minimization. Addressing this possibility is important and this was precisely the reason why we discussed in Section 4 why the trivial solution does not confound our results. To recap, our argument is as follows: the trivial solution of assigning all nodes to the same embedding never occurs in our experiments, since if such a representation were to have emerged, it would destroy all information about the current token and hence we would not find (i) different representations for different tokens in the PCA projections, nor (ii) high accuracy of solving our graph tracing tasks. We also note that as long as this trivial solution is ruled out, Theorem 5.1 shows that minimizing the energy will give us the PCA results we present in the paper: this rigorously shows our results are not confounded by degenerate representations.
>
> Having said the above, an alternative way of addressing the possibility of a trivial solution is by redefining the energy metric to render such a solution infeasible. Specifically, we can standardize the activations (i.e., mean-center them and divide by standard deviation) before performing the energy calculation. This operation will assign infinite energy to the trivial solution, since assigning all concepts the same representation will yield a standard-deviation of zero, and hence an infinitely large energy value. We have added new results with this standardized energy metric in Appendix C.3 (Figure 13). **The results corroborate our claims above, that the representations are not degenerate** and show qualitatively the same trends as the non-standardized energy metric reported in the main paper. However, for the results on ring graphs, there are fluctuations in the energy values as a function of context size. *This is expected*, since standardization also induces extreme sensitivity to noisy dimensions, which can yield a lot of fluctuations in plots. For this reason, we still report the original energy metric in the main paper, but include the standardized metric in the appendix to guarantee the reader that the trivial solution does not confound our results.
>
> ---
>
> > **While I don't count this as a significant weakness, the paper could arguably be improved by further exploring what in the model causes the abrupt shift in representations. (e.g. is there any connection with induction heads, transformer circuits, or any other mechanism that could be responsible for the change).**
>
> We certainly agree! Currently, we have tried to visualize attention patterns, but without knowing which heads are causal, it was hard to identify the mechanism driving the transition. We hope to investigate the mechanism driving the shift in representations by, e.g., attribution patching as in [1]; however, we believe this investigation is likely to be a rich topic in itself, and hence better suited for a future study.
>
> [1] https://arxiv.org/pdf/2310.15213
>
> ---
> ---
>
> **[Continued below...]**

---

> ### Author Response · Authors · 2024-11-22
> **Rebuttals (2/2)**
>
> **Further Questions.**
>
> > **Authors found that there is a shift in concept representations after the model has seen enough context. Could this mean that the model does not 'understand' early examples in real few-shot learning tasks? ...**
>
> This is an interesting question! While it is difficult to provide a confident answer without running experiments, we believe that most real few-shot learning tasks operate in the regime of “task-retrieval”, wherein the in-context exemplars are merely used to specify which capability acquired by the model during pretraining is to be used on the provided in-context exemplars. For such tasks, it is usually the case that the model constructs a task / function vector [1, 2], and there isn't much accuracy gain once this vector is formed and hence the task is easily retrieved. For example, one generally observes rather quick saturation of accuracy (e.g., in 3–5 exemplars). For these reasons, we believe generalizing our claims to popular few-shot learning tasks is likely to be difficult. However, it is likely that novel tasks that are unlikely to be seen by the model during pretraining (e.g., similar to the task proposed by Pan et al. [3] or our graph tracing tasks) may be good candidates for investigating the hypothesis put forth by the reviewer. These experiments are currently outside the scope of this paper, but we believe they will be interesting to explore in a future work.
>
> [1] https://arxiv.org/pdf/2310.15213
>
> [2] https://arxiv.org/abs/2310.15916
>
> [3] https://arxiv.org/abs/2305.09731
>
> ----
>
> > **What happens to the representation shift if you increase the number of concepts? Is there a point by which the model is no longer able to shift the structure properly to match the in-context 'meanings'?**
>
> Our current experiments scale up to **1000 concepts**, and we find that the model is able to perform our proposed task and show the transitions seen in Figure 8.a up to this scale. However, going beyond a 1000 concepts has been difficult due to memory constraints---the more concepts we have, the longer the context length needs to be to retrieve our results, hence bottlenecking us.
>
>
> ----
> ----
> **Minor points.**
>
> > **L186 task-specific? (more popular with a dash)**
>
> Thank you, we have corrected this everywhere.
>
> > **L269 “Given that we empirically do not observe this to be the case, we can safely assume this trivial solution does not arise in our experiments.” - slightly odd phrasing - you make an observation about your experiments that implies an assumption about the same experiments?**
>
> This phrase was indeed awkwardly worded and we have updated it now (please see Section 4). We now discuss more precisely why our experiments from other sections guarantee the trivial solution does not confound our energy computation results. We also note, as mentioned in our responses above, that we have added a new experiment where we use standardized representations in energy computation to ensure the trivial solution is infeasible, finding qualitatively similar results (Appendix C.3). We hope these changes help address the reviewer's concerns!
>
> ----
> ----
>
> **Summary.** We thank the reviewer for their feedback which motivated us to check our claims on different models and improve our energy metric! Specifically, we have now added experiments on several model families (Llama3.2-1B, Gemma2-2B, Gemma2-9B, and Llama3.1-8B-Instruct), expanded our discussion on why trivial solution does not affect our energy computation results, and added new results with standardized representations in Appendix C.3. We hope our responses properly address the reviewers comments properly and we would be grateful if they can champion our paper for acceptance!

---

> > ### Author Response · Authors · 2024-11-24
> >
> > Dear Reviewer,
> >
> > We thank you again for your detailed feedback on our work. Given the discussion period ends soon, we wanted to check in if our provided responses address your concerns, and see if there are any further questions that we can help address.
> >
> > Thanks!

---

### Author Response · Authors · 2024-11-22
**General Rebuttal**

Dear Reviewers,

We would like to thank you for your insightful feedback! We are pleased that the reviewers unanimously found our novel synthetic data generation process and the observation of in-context emergence of structured representations to be: “interesting phenomena” (R. KJBF), “experimental design is original, elegant and well crafted” (R. a5tz), “the experiment results are interesting” (R. EbvH), and “the set of experiments are spot-on” (R. hdrb). We also appreciate the feedback and constructive suggestions for validating the robustness of our observations, which motivated us to run an extensive set of new experiments across different combinations of large language models (LLMs), graph structures, and energy functions of interest. We are glad to report that our results hold robustly across these conditions!

To fully reflect your feedback, we have implemented the following updates in our revised manuscript:

**New Experimental Results:**


- **E1: Validation of claims on additional models (Figs. 11, 12, 13).** We thank R. KJBF and hdrb for highlighting the importance of verifying our claims on additional models to enhance the robustness of our conclusions. In response, we conducted experiments on four additional models (Llama3.2-1B, Gemma2-2B, Gemma2-9B, and Llama3.1-8B-Instruct) across three distinct graph structures (square grid, ring, and hexagonal grid).  The 42 new plots resulting from these experiments are presented in Figures 11, 12, and 13 in App. C.2 and C.3. Our qualitative findings from the PCA visualizations remain consistent (Fig. 11).


- **E2: Representations are not degenerate**: We confirm that the representations are not degenerate by demonstrating that the Dirichlet energy results hold after both normalization (Fig. 12) and standardization (Fig. 13). This resolves R. KJBF’s concern that all nodes might be assigned to the same embedding, thereby reinforcing our original claims.


- **E3: The PCA dimensions are causally relevant (Table 1).**  We conducted experiments demonstrating that the top principal components encode information causally relevant to how our model solves the task. Specifically, in App. C.4, we performed interventions on the representation projected onto top PCs of the last token presented to the model, showing that we can manipulate the model's belief about its current node in the graph, regardless of the actual last token. This demonstrates that the principal components not only represent the graphical structures but also capture causally relevant information used by the model to solve our in-context learning tasks.


- **E4: Quantifying similarity of PCA components and spectral embeddings (Table 2).** In response to the comments from R. a5tz, EbvH, and hdrb, we have quantified the cosine similarity between the graph's spectral embeddings and the principal components identified in our study. This analysis, presented in Table 2 of App. C.5, further reinforces the connection between our empirical results and spectral graph theory.


- **E5: Energy quantification for the semantic prior experiment (Fig. 14).** In response to R. EbvH's comment, we have strengthened the argument made by Fig. 3 by adding new analyses in Fig. 15. In addition to clarifying the interpretations of the results in Fig. 3c, we have quantified the energy in the first and second principal components, which are believed to embed the semantic structure. And we have also quantified the energy in the third and fourth principal components, which appear to support the in-context structure. As expected, we found that the energy of the dimensions supporting the in-context structure decreases as more context is added.

**Manuscript Updates:**


- **M1: Strengthening motivation for our synthetic task (Sec. 2).** We have enhanced the motivation for our synthetic in-context learning task by discussing recent studies on LLM representations, the use of LLMs for in-context reinforcement learning tasks, and existing experiments on humans. In Section 2, we have better emphasized how our work relates to existing literature to provide stronger grounding and motivation for our study.


- **M2: Clarifying the use of PCA and Dirichlet energy (Sec. 3,4).**  R. KJBF and EbvH highlighted the subtlety of using PCA to measure structure and draw conclusions. Initially, we used PCA to visually present results and motivate further validation through the computation of Dirichlet energy. Furthermore, we have theoretically demonstrated how spectral graph theory motivates PCA to faithfully embed spectral embeddings of the graph, as verified in E4 above. The reviewers' feedback prompted us to better emphasize this logical connection. To this end, we have updated Section 3.1 to assure readers that proper quantification of structure will follow in Section 4.

With these substantial updates, we believe our revised manuscript now addresses all the concerns. Please let us know if you have any further questions!

---

### Meta-Review · Area_Chair_oBn7 · 2024-12-29

**Metareview:**

This paper investigates how LLMs can reorganize their internal representations during in-context learning to reflect novel task-specific structures, rather than being constrained by semantic relationships learned during pre-training. The authors demonstrate this through a "graph tracing" task where models must learn to navigate graphs whose nodes are labeled with common words (like "apple", "bird") but connected in arbitrary ways. The authors show that with sufficient context examples, models can suddenly reorganize their internal representations to mirror the graph structure rather than semantic relationships, as evidenced through PCA visualizations, Dirichlet energy measurements, and causal intervention experiments. The reviewers appreciated the experimental design of the work, thorough experimental validation on multiple models and graph types, as well as theoretical grounding in graph theory and energy minimization. The weaknesses were limited experimentation on Llama, reliance on PCA, and the synthetic nature of the task. Some of these were addressed during the rebuttal/discussion phase. Overall, the reviewers were in broad agreement to accept the work.

**Additional Comments On Reviewer Discussion:**

See above.

---

### Decision · Program_Chairs · 2025-01-22

Accept (Poster)